# Leveraging Color Channel Independence for Improved Unsupervised Object Detection

## Abstract

Object-centric architectures can learn to extract distinct object representations from visual scenes, enabling downstream applications on the object level. Similarly to autoencoder-based image models, object-centric approaches have been trained on the unsupervised reconstruction loss of images encoded by RGB color spaces. In our work, we challenge the common assumption that RGB images are the optimal color space for unsupervised learning in computer vision. We discuss conceptually and empirically that other color spaces, such as HSV, bear essential characteristics for object-centric representation learning, like robustness to lighting conditions. We further show that models improve when requiring them to predict additional color channels. Specifically, we propose to transform the predicted targets to the RGB-S space, which extends RGB with HSV's saturation component and leads to markedly better reconstruction and disentanglement for five common evaluation datasets. The use of composite color spaces can be implemented with basically no computational overhead, is agnostic of the models' architecture, and is universally applicable across a wide range of visual computing tasks and training types. The findings of our approach encourage additional investigations in computer vision tasks beyond object-centric learning.

## 1 Introduction

The ability to form abstract, structured representations of the physical world is a cornerstone of human intelligence (Lake et al., 2017; Fodor & Pylyshyn, 1988). It enables us to excel in dealing with out-of-distribution situations, reasoning, analogy-making, and causal inference. Similarly, machine learning models that adapt the structural knowledge of an environment are more efficient and generalizable (Schölkopf et al., 2021). Modeling language as a sequence of structured tokens was a significant step toward the applicability and generalization of LLMs (Sennrich et al., 2016; Devlin et al., 2019). Similarly, many computer vision applications might also profit from segmenting *visual scenes* into *semantic* tokens, but inducing those is less self-evident. Thus, state-of-the-art models primarily represent images as a whole or as fixed-sized grids, consequently failing at simple reasoning that requires structural knowledge of physical entities such as counting objects (Radford et al., 2021).

Addressing this issue, *object-centric representation learning* (OCRL) aims to infer a set of slots, each representing a distinct object - allowing for causal inference *at the object level* (Ding et al., 2021). Although binding an object to a slot from raw perceptual input is challenging (Greff et al., 2020a), recent advancements in OCRL, such as the Slot Attention (SA) module (Locatello et al., 2020), enabled unsupervised learning for simple synthetic datasets. However, scaling these methods to complex datasets proves challenging - slots attend to features considered irrelevant (Kipf et al., 2020) such as static backgrounds, represent spatial areas instead of objects (Seitzer et al., 2023), and are highly entangled (Singh et al., 2023). Subsequently, researchers investigated modified architectures utilizing the compositional nature of objects (Biza et al., 2023; Singh et al., 2023), optimization methods respecting the generative process of visual scenes (Brady et al., 2023; Wiedemer et al., 2023), and (semi-)supervised objectives (Elsayed et al., 2022b; Seitzer et al., 2023). Substantial performance increases of the latter highlighted that a pure RGB reconstruction loss may be a signal that is too weak for slots to segment scenes (Seitzer et al., 2023). Our work follows this assumption - However, we provide additional *unsupervised signals by augmenting the RGB color space with color channels from other color spaces*. We reason about our research in the following.

As in many vision-based approaches, the training data for unsupervised OCRL are color images, represented as red, green, and blue components, i.e., the classical RGB color space. To the best of our knowledge, all previous work for OCRL trains models on such RGB scenes. The RGB representation can be a sensible choice for images, primarily due to convenience in vector format mapping, its byte size, and its similarity to the human eye apparatus (Podpora et al., 2014). However, we argue that the RGB space is insufficient for unsupervised OCRL due to its color channels' high correlation, sensitivity to lightness, and non-uniformity. Other color spaces (which are *non-linear* and *non-continuous* transformations from RGB), such as HSV, do not suffer from these problems. Furthermore, they include highly discriminative features for object detection (see Figure 1) However, they introduce other drawbacks detailed in section 2. Thus, the question arises of how OCRL models can profit from the strengths of various color spaces while not inheriting their weaknesses - we propose to solve this problem by developing *novel composite color spaces that combine expressive color channels from multiple spaces*. We argue and show empirically that those color channels that are robust to some effects, such as lighting or change in saturation, significantly improve the performance of OCRL models.

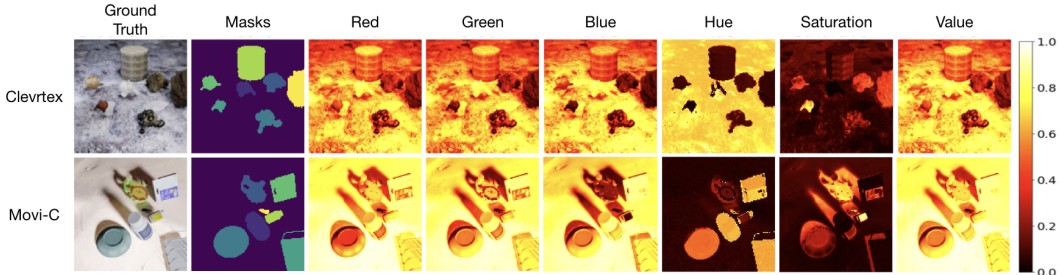

Figure 1: Exemplary scene from the Clevrtex and Movi-C dataset. We plot the original image and object masks with individual color channels on a heatmap. We show RGB's Red, Green, and Blue channels and HSV's Hue, Saturation, and Value. The objects are distinguishable from the background and other objects in all heatmaps. Contrary to the RGB channels, the HSV channels are uncorrelated.

**Our main contributions are:**

- We claim that RGB image representations are suboptimal for unsupervised OCRL as their color channels are *highly correlated, sensitive to light effects, and non-uniform* for natural images.

- We show that the color space representation of the *target* significantly influences the performance of Slot Attention models and discuss the strengths and drawbacks of color channels of multiple well-known color spaces. Consequently, we create novel, composite color spaces by combining complementary color channels. Previous work only focuses on the color space representation of the *input*.

- We demonstrate that the augmented color spaces are *model-agnostic, generalizable, and efficient while significantly outperforming* the established RGB color space for object detection and property disentanglement for the five multi-object datasets Clevr (Johnson et al., 2016), Multishapenet 4, Multishapenet 24 (Stelzner et al., 2021), Clevrtex (Karazija et al., 2021), and Movi-C (Greff et al., 2022).

## 2 BACKGROUND: SLOT ATTENTION

In the following, we introduce the technical background for Slot Attention (SA) (Locatello et al., 2020). We use SA for all our experiments, but our composite color spaces are transferable to other OCRL implementations.

Traditional encoder-decoder architectures for computer vision generate a single, fixed-size (high-dimensional) latent vector $\mathbf{z}$ for an observation $\mathbf{x}$. The latent space should accurately represent the underlying data distribution and capture the generative factors (Bengio et al., 2013; Locatello et al.,

2019). In OCRL, we assume data consists of individual objects. Thus, object-centric architectures aim to induce a latent space decomposed into distinct objects.

For example, SA is an iterative algorithm employed on the learned features of an encoder. Each iteration starts with $K$ `slots` $\in \mathbb{R}^{K \times D_{slots}}$ and `inputs` $\in \mathbb{R}^{N \times D_{enc}}$, the position-embedded outputs of an encoder. $K$ must be user-defined or estimated. First, attention values $\texttt{attn}_{i,j}$ are computed between each `slot` and the `inputs` using dot-product attention with learnable linear queries and keys (Vaswani et al., 2017). Crucially, SA introduces competition by softmax normalizing the attention coefficients over the `slots`, distributing `inputs` information among the `slots`.

After introducing competition, attention weights `attn` and a learnable function $v$ embed the values to update the `slots`. Finally, a Gated Recurrent Unit and another MLP merge the `updates` into the `slots`. For the mathematical details and formulas, we refer to Locatello et al. (2020).

The SA algorithm delivers `slots` that distributedly store the *input* information. Depending on the task and data, learned slot representations can be used in different downstream architectures. For unsupervised object discovery, which we examine in our work, slots are spatially broadcasted and decoded separately with a weight-shared decoder - the decoder usually predicts a *four-dimensional output, capturing the RGB representations and masks*. Masks are softmax normalized to merge individual reconstructions into a combined output (Greff et al., 2020b; Locatello et al., 2020).

Because RGB images strongly correlate with the lightness (see "value" in Table 7) and the models are optimized with a simple pixel-wise reconstruction loss, models predominantly focus on predicting the lightness accurately. Consequently, the binding mechanism discriminates spatial areas primarily by a single dimension. Other perceptual variables, such as the hue and saturation (Smith, 1978; Schwarz et al., 1987) that may be important to discriminate between objects are thus underrepresented. We argue that requiring models to additionally decode complementary color channels, such as hue and saturation, provides a stronger unsupervised signal for segmenting semantic objects.

## 3  COLOR SPACES FOR UNSUPERVISED OBJECT CENTRIC REPRESENTATION LEARNING

In this section, we highlight the impact of color choices on unsupervised OCRL. Outgoing from the common RGB color space, we discuss requirements for sensible color representations and arrive at composite color spaces that combine channels with complementary information of multiple spaces.

### 3.1  WHY IS RGB A SUBOPTIMAL CHOICE FOR NATURAL SCENES IN OCRL?

We argue that RGB is a suboptimal color space for unsupervised OCRL, as RGB is

- **strongly correlated**: The RGB color space consists of the additive "Red," "Green," and "Blue" color channels. Those are all strongly correlated in natural images. We provide empirical evidence for that in Table 7, where we calculated the Pearson Correlation Coefficient between each color channel for 1000 images, respectively sampled from *ClevrTex* (Karazija et al., 2021), *MS Coco* (Lin et al., 2014), and *ImageNet* (ILSVRC2012) (Deng et al., 2009). The correlation not only introduces redundancy into machine learning models but clashes with the perceptual variables of humans (Smith, 1978; Schwarz et al., 1987): As the RGB space is optimized with a reconstruction loss, models primarily focus on predicting the value (or lightness). Contrarily, humans' color vision assimilates incoming light sources differently — while RGB cones produce the first physical stimulus, the stimuli are transformed into the *sensations* of hue, colorfulness ("saturation"), and brightness ("value") (Müller & , Lipsk; Kim et al., 2009). Our brain finally uses those sensations to discriminate between objects. Consequently, objects that are similar in the perception of humans (but differ in some properties, e.g., their hue) are scattered by the RGB space.

- **sensitive to lighting**: The RGB channels are not only strongly correlated with each other but also with the value/lightness (see Table 7). In natural scenes, the same objects might be represented by entirely different RGB color histograms due to rapid changes in lighting. We elaborate on this effect in subsection D.4 and even show that the slot representations in the HSV space are less dependent on the lighting conditions.

- **non-uniform in the perceived color space**: Distances in uniform color spaces reflect the perceived distance of humans (Paschos, 2001). The RGB space is non-uniform, leading to distorted representations of visually similar features. As slots attend to similar features (based on color representations), the binding mechanism might be negatively influenced.

## 3.2 ARE OTHER COLOR SPACES BETTER SUITED FOR UNSUPERVISED OBJECT CENTRIC REPRESENTATION LEARNING?

Other color spaces, such as HSV and CIE-LAB solve some of the problems discussed for the RGB space. HSV is a cylindrical-coordinate representation of RGB, modeling the "Hue," "Saturation," and "Value." The value strongly correlates with the RGB channels (see Table 7). Contrarily, the hue and saturation are not correlated to the RGB channels and are only slightly influenced by lighting. Similarly, it aligns with the perceptual variables of humans (Smith, 1978; Schwarz et al., 1987). Furthermore, it is almost perceptually uniform. CIELAB models the luminosity along with the two color dimensions. Due to the explicit modeling of luminosity, it is perceptually uniform, non-correlated, and robust to light influences. HSV and CIE-LAB can be losslessly transformed from the RGB space. Although those transformations are non-continuous, they can be approximated by machine learning algorithms. However, we argue and empirically verify that transformations in the *target* representations significantly influence OCRL models' performance. As the models are trained on the pixel-wise reconstruction error, models may use their predictive capacity differently depending on the color space. For example, if we use RGB space, we rely on the OCRL model to implicitly learn objects' invariances due to lightness effects while we model them in HSV with the hue and saturation component. We verify in subsection D.4 that the robustness of the HSV channels to different lighting conditions transfers to the slot representations - objects that share the same underlying factors remain more similar when confronted with different lighting.

On the contrary, HSV and CIE-LAB introduce other problems. For example, HSV's hue is discontinuous and thus not well modeled by differentiable machine learning algorithms, and transformations from CIE-LAB to RGB are not lossless. Thus, our work doesn't rely on a single color space but later creates composite color spaces. To identify which color channels are suitable for composite spaces, we first assess which color spaces impact OCRL's performance. Therefore, we trained SA models with the same architecture and hyperparameters predicting RGB, CIE-LAB, or HSV scenes on the Clevr and Clevrtex datasets. The input scenes are always represented as RGB scenes, and we trained on six random seeds. Machine learning algorithms can approximate transformations of the inputs and thus have less impact on the performance (Mishkin et al., 2017). We further discuss input transformations in subsection D.3. We report the Foreground Adjusted Rand Index in Table 1, measuring object discovery in scenes.

Table 1: Object Discovery FG-ARI (↑) for various color spaces (RGB2X)

| Dataset | RGB-RGB | RGB-LAB | RGB-HSV | RGB-GRAY | RGB-R |
|---|---|---|---|---|---|
| Clevr | $94.1 \pm 1.1$ | $95.0 \pm 1.0$ | $43.8 \pm 9.3$ | $87.6 \pm 1.3$ | $92.6 \pm 1.5$ |
| ClevrTex | $71.7 \pm 1.4$ | $73.1 \pm 8.5$ | $86.7 \pm 1.7$ | $63.6 \pm 5.5$ | $61.8 \pm 4.0$ |

RGB and CIELAB perform similarly. They achieve a close-to-perfect segmentation on the simple Clevr dataset but perform worse when including photorealistic textures in Clevrtex. Although we argued that RGB is highly correlated, it still performs significantly better than grayscale values. However, even the single gray dimension suffices to segment most scenes for the Clevr dataset. Similarly, training only on the R dimension suffices to segment scenes on Clevr but lacks performance on Clevrtex. We make an interesting observation for the HSV space: On the one hand, models degenerate to bind *areas* instead of objects for Clevr. We suspect that the discontinuity in the hue channel (see Figure 2, "Hue" and "Masks") distorts the binding mechanism. On the other hand, models trained on HSV show a significantly improved performance for Clevrtex. We suspect that the fine-grained differences between objects and backgrounds are more explicitly modeled in HSV (see Figure 1) - as the RGB space strongly correlates with the value dimension of HSV, the hue and saturation are underrepresented. Compared to RGB, slots are thus provided with more information to discriminate between objects. The discontinuity in the hue channels is less influential as

SA models do not group object based on their hue alone due to complex textures and illumination effects.

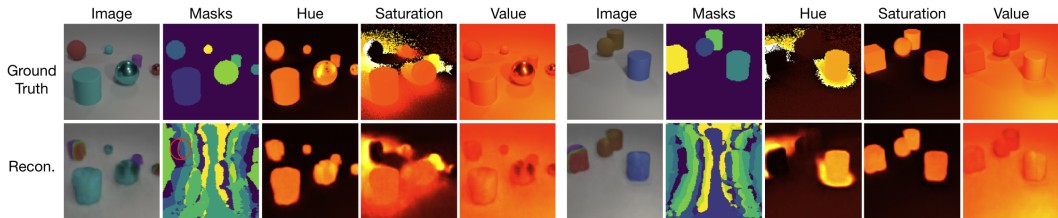

Figure 2: Two qualitative samples for SA with a RGB-HSV reconstruction on Clevr. We observe that the reconstruction and slot assignment are degenerated, with scenes split into many different stripe-like slots. For models trained on RGB, RGB-S, RGB-SV, the mask reconstruction almost perfectly matches the ground-truth mask (for more details, see Figure 9)

### 3.3 COMBINING COLOR SPACES

While we previously gauged between individual color spaces, we will now argue about combining complementary color channels into a composite target space. Combined color spaces have been initially considered in the literature but have yet to be focused on OCRL (Du et al., 2024). Models trained on composite color spaces are less dependent on inherent biases of the color spaces. For example, differences in hue and saturation are subtle in the RGB space (due to the channels high correlation with the lightness) but explicitly incorporated in the HSV space. For the following paper, we mainly use the stability and reliability of the RGB space and enrich it with complementary channels of the HSV space - especially the saturation bears interesting characteristics for unsupervised OCRL due to its non-correlation to the RGB channels, its continuity, and robustness against lightness. We show that this single dimension already *significantly* improves the performance of object detection and intra-object disentanglement (see Table 5.1.1 and Figure 4). Moreover, we tested further combinations of RGB and HSV, but their improvements were less significant or unstable.

The SA model requires minimal change in architecture for the enhanced color spaces. We solely alter the spatial broadcast decoder to predict a five-dimensional output for the RGB-S space (instead of four) consisting of the RGB channels, the saturation, and the alpha masks. The parity of the alpha masks among the four color channels reflects the consistency of the color channels for these objects: *Slots are required to model two independent but semantically relevant targets that are spatially compact*. Similar to the minor adjustments in the architecture, the loss function remains an unsupervised reconstruction loss but includes saturation (or any other channel) as a target. For our experiments, the input to the networks remains RGB. Although we argue that complementary color channels are helpful as signals in the target, we estimate that their influence is less substantial for inputs: The RGB color space already contains all information, and the encoder can learn the transformations. However, we consider the exploration of the input space an interesting future work, although we report mixed initial results (see Table 8).

## 4 RELATED WORK

To our knowledge, combined color spaces have not yet been explored for object-centric representation, although they have shown promising success in computer vision tasks (Shin et al., 2002; Gowda & Yuan, 2019; Li et al., 2011). However, they focus on transformations on the *input* instead of the *targets*. In particular, using different color spaces such as HSV or YCbCr were investigated for challenging segmentation and detection tasks (Shin et al., 2002; Vandenbroucke et al., 2003; Taipalmaa et al., 2020). The choice of input color space might have been a typical approach in feature engineering. However, with the advent of deep learning-based methods, such feature engineering has lost its importance in favor of simple and general methods (Mishkin et al., 2017). Existing work has focused on general color transformations Miao & Wang (2023), the usage of a single alternate color space (Lim et al., 2020), or colorspace conversion within autoencoders (Akbarinia & Gil-Rodríguez, 2021). All have shown mixed results, primarily due to the usage of whole color spaces. Only very recently, the use of composite color spaces (Tan et al., 2023) has been investigated in deep learning,

albeit for image colorization (Du et al., 2024). We revisit color space transformations for *unsupervised* OCRL. The challenging nature of this task and the transformation of the output (instead of the input) makes our work conceptually different from all presented work.

The observation that the RGB pixel-wise reconstruction loss does not suffice to bind semantic objects in natural scenes is discussed in the literature (Seitzer et al., 2023; Kipf et al., 2022). Consequently, Elsayed et al. (2022a) enhance the RGB output space with a supervised depth signal for object discovery in videos. They build their work on Kipf et al. (2022) which find that SA for videos strongly improves when conditioning slots on objects in the first frame. Similarly, Kim et al. (2023) utilize point-based descriptors to prohibit slots to focus on backgrounds. While all of the above works utilize pixel-wise reconstruction losses, Seitzer et al. (2023); Qian et al. (2023); Wen et al. (2024); Aydemir et al. (2023) abstain from training slots on a reconstruction error and instead predict semantic semi-supervised signals. Similarly, Xu et al. (2022) leverage embedded text descriptions as supervision for semantic segmentation of images. Another unsupervised alternative is discussed by Brady et al. (2023) and Wiedemer et al. (2023) - they leverage derivatives in the loss function to explicitly model the compositionality of OCRL scenes. In our evaluation, we also test whether the additional color channels influence the representation of underlying generating factors in slots. Biza et al. (2023) explicitly disentangle position, size, and pose information by invariant position embeddings. Majellaro et al. (2024) extend this framework to disentangle the shape and texture of objects. Singh et al. (2023) introduce the concept of blocks that bind different *factors* of objects, and Stammer et al. (2024) discretize those blocks. Combined color spaces can be used complementary to those approaches.

## 5 EXPERIMENTS AND RESULTS

For our experiments, we investigate the impact of five color spaces on unsupervised Slot Attention. Our baselines are RGB and HSV, and we utilize RGB-S, RGB-SV, and RGB-HSV as combined color spaces. For comparability, all models use the same architecture; only the final output layer is adjusted to match the dimension of the color spaces. As motivated in subsection 3.2, we only transform the *targets*, and the input remains RGB.

We mainly evaluate object discovery, measuring whether slots segment scenes into distinct objects. In line with previous work, we report the Foreground-Adjusted Rand Index (FG-ARI) and the mean Intersection over Union (mIoU). Additionally, we report the RGB-MSE for scene reconstruction quality. Note that if a network predicts additional color channels, they are *not* considered for the MSE measurement, and HSV is algorithmically transformed to RGB for this to work. Furthermore, we evaluate how well the slot representations generalize to the underlying generating factors of objects. We utilize the DCI framework (Eastwood & Williams, 2018) in line with Singh et al. (2022). Previous work investigated that unsupervised SA trained on RGB mainly focuses on low-level features such as color statistics (Seitzer et al., 2023) - we show that one additional, uncorrelated color channel already suffices to achieve a significantly higher generalization of slots. We evaluate the color spaces on five multi-object datasets, namely Clevr, MultiShapeNet (4 and 24), Clevrtex, and Movi-C (Johnson et al., 2016; Stelzner et al., 2021; Karazija et al., 2021; Greff et al., 2022). We treat Movi-C as an image dataset. Model and optimization parameters are detailed in Appendix A. Furthermore, we provide qualitative results in Appendix D.

### 5.1 RESULTS ON OBJECT DISCOVERY

We measure object discovery for the five datasets Clevr, MultiShapeNet (4 and 24), Clevrtex, and Movi-C (Johnson et al., 2016; Stelzner et al., 2021; Karazija et al., 2021; Greff et al., 2022). An overview of the results is given in Table 5.1.1, and we detail the results in the following sections.

### 5.1.1 CLEVR

We start our evaluation with the Clevr dataset (Johnson et al., 2016), containing at most ten simple geometric objects per scene on a gray background. For our evaluation, we use the model architecture introduced by Locatello et al. (2020), achieving near-perfect segmentations. We already discussed in section 3 that models trained on the HSV degenerate. We argued that the discontinuity of the hue channel prohibits slots from binding objects. Thus, as an initial proof of concept, we first verify

that the enhanced RGB color spaces, without non-continuous hue, do not degenerate. We report segmentation and reconstruction quality for all datasets in Table 5.1.1. Results of models trained on RGB values are consistent with the literature (Locatello et al., 2020). While models trained with the Hue component degenerate (visible in the segmentation and reconstruction quality), models trained on the RGB-S and RGB-SV space do not degenerate and even improve the segmentation quality slightly. However, the combined color spaces do not help segment objects from the background. Thus, the mIoU score is low for all models.

Table 2: Evaluation results on five datasets for the different predicted output color spaces. Best values are bold (multiple if in same error band). Best predictions indicated in bold if consistent lead for both FG-ARI and mIoU. It can be seen that, overall, enhancing RGB with other color dimensions leads to strong performance improvements. Primarily RGB-S and RGB-SV lead to a *consistent* performance improvement.

| | V | Pred. | FG-ARI ↑ | mIoU ↑ | RGB-MSE* ↓ |
|---|---|---|---|---|---|
| *Clevr* | CNN | RGB | $94.1 \pm 1.1$ | $\mathbf{26.9 \pm 0.5}$ | $54.0 \pm 5.1$ |
| | | **RGB-S** | $\mathbf{94.9 \pm 1.2}$ | $\mathbf{27.0 \pm 2.0}$ | $\mathbf{46.5 \pm 3.9}$ |
| | | **RGB-SV** | $\mathbf{95.4 \pm 0.5}$ | $\mathbf{26.7 \pm 0.7}$ | $\mathbf{46.0 \pm 1.4}$ |
| | | RGB-HSV | $88.3 \pm 13.2$ | $\mathbf{25.6 \pm 1.8}$ | $51.4 \pm 5.5$ |
| | | HSV | $44.9 \pm 8.0$ | $17.3 \pm 2.4$ | $99.8 \pm 9.5$ |
| *MultiShapeNet-4* | CNN | RGB | $73.7 \pm 11.8$ | $26.9 \pm 17.5$ | $76.8 \pm 13.2$ |
| | | **RGB-S** | $\mathbf{82.1 \pm 2.5}$ | $\mathbf{55.0 \pm 16.8}$ | $73.5 \pm 6.5$ |
| | | **RGB-SV** | $\mathbf{81.9 \pm 7.5}$ | $\mathbf{56.2 \pm 14.3}$ | $\mathbf{60.3 \pm 5.4}$ |
| | | RGB-HSV | $62.7 \pm 20.2$ | $21.6 \pm 9.5$ | $87.1 \pm 5.6$ |
| | | HSV | $61.1 \pm 9.7$ | $15.6 \pm 3.3$ | $116.9 \pm 6.3$ |
| *MultiShapeNet-24* | CNN | RGB | $61.6 \pm 13.2$ | $\mathbf{33.6 \pm 21.5}$ | $\mathbf{45.6 \pm 5.0}$ |
| | | **RGB-S** | $\mathbf{70.3 \pm 4.7}$ | $\mathbf{46.6 \pm 19.4}$ | $\mathbf{44.8 \pm 5.2}$ |
| | | **RGB-SV** | $\mathbf{71.7 \pm 6.3}$ | $\mathbf{43.5 \pm 19.8}$ | $\mathbf{44.5 \pm 4.3}$ |
| | | RGB-HSV | $62.0 \pm 12.6$ | $22.1 \pm 11.6$ | $53.4 \pm 6.0$ |
| | | HSV | $57.7 \pm 12.6$ | $22.8 \pm 11.8$ | $74.1 \pm 8.1$ |
| *Clevrtex* | CNN | RGB | $66.3 \pm 16.3$ | $41.5 \pm 13.0$ | $\mathbf{293.4 \pm 13.1}$ |
| | | RGB-S | $\mathbf{84.2 \pm 3.6}$ | $61.5 \pm 3.5$ | $320.1 \pm 8.1$ |
| | | RGB-SV | $81.7 \pm 3.8$ | $53.3 \pm 7.8$ | $308.6 \pm 8.8$ |
| | | RGB-HSV | $\mathbf{85.8 \pm 1.9}$ | $64.2 \pm 4.4$ | $318.2 \pm 9.4$ |
| | | **HSV** | $\mathbf{86.4 \pm 1.7}$ | $\mathbf{70.0 \pm 2.0}$ | $371.9 \pm 10.2$ |
| | ResNet | RGB | $75.6 \pm 12.0$ | $48.6 \pm 14.8$ | $\mathbf{179.9 \pm 5.4}$ |
| | | RGB-S | $\mathbf{92.7 \pm 2.4}$ | $73.6 \pm 3.2$ | $192.1 \pm 11.0$ |
| | | RGB-SV | $88.7 \pm 10.3$ | $67.2 \pm 14.2$ | $211.5 \pm 17.5$ |
| | | RGB-HSV | $88.6 \pm 9.8$ | $66.7 \pm 13.8$ | $195.9 \pm 5.2$ |
| | | **HSV** | $\mathbf{94.8 \pm 0.8}$ | $\mathbf{81.1 \pm 1.1}$ | $244.3 \pm 4.3$ |
| *Movi-C* | ResNet | RGB | $42.9 \pm 3.3$ | $21.3 \pm 1.8$ | $\mathbf{192.4 \pm 139.5}$ |
| | | **RGB-S** | $\mathbf{45.7 \pm 2.4}$ | $\mathbf{27.2 \pm 2.7}$ | $\mathbf{211.2 \pm 96.1}$ |
| | | RGB-SV | $\mathbf{43.2 \pm 4.9}$ | $22.3 \pm 4.3$ | $\mathbf{251.5 \pm 173.8}$ |
| | | RGB-HSV | $40.7 \pm 3.0$ | $\mathbf{29.0 \pm 3.5}$ | $425.3 \pm 124.7$ |
| | | HSV | $38.1 \pm 6.0$ | $24.8 \pm 6.6$ | $626.7 \pm 71.9$ |

*\* MSE caveat: A tendency for higher values for the enriched color spaces and HSV is expected as the MSE involves a conversion to RGB.*

### 5.1.2 MULTISHAPENET

We consider two variants of Multishapenet (Stelzner et al., 2021), which we call "4" and "24". In the `4` version, we filtered for images containing *exactly* four objects, while we allowed 2 to 4 objects in the `24` version. We follow the translation and scaling invariant architecture by Biza et al. (2023). Similarly to Clevr, incorporating the hue channel leads to model degeneration. Conversely, adding Saturation to the RGB color space significantly improves segmentation quality for both dataset variants. For MSN `4`, FG-ARI is improved from 73.7 to 82.1, and for MSN `24`, from 61.6 to 70.3, representing a relative improvement of 10%. The differences are even more substantial for the mIoU score: Although the RGB models often succeed in distributing different objects to different

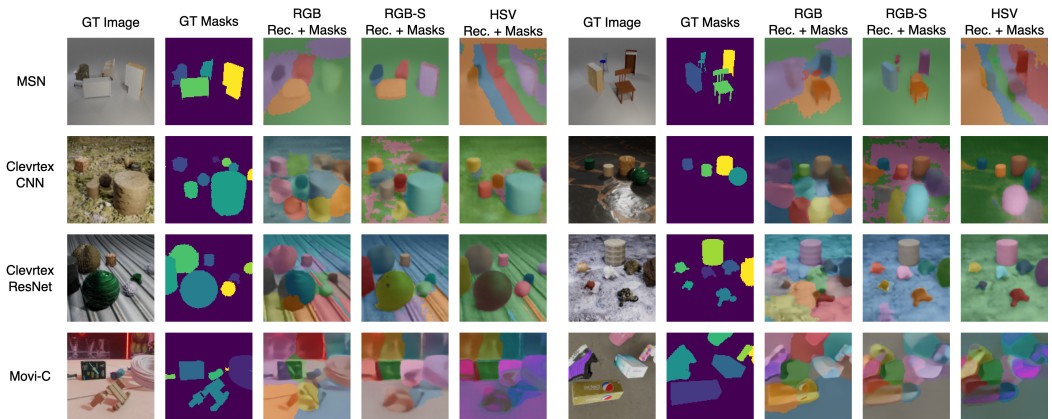

Figure 3: We show reconstructions and masks of models trained on Multishapenet, Clevrtex, and Movi-C compared to their ground truth. While the RGB space does not confidently segment objects from each other and from the background, RGB-S achieves close-to-perfect scene segmentations. On Movi-C, RGB models degenerate to represent spatial areas instead of objects. The RGB-S slots mostly attend to semantic objects, but objects are often split into multiple slots.

slots, they do not segment them clearly from the background. We visualize that behavior in Figure 3 (top row). We made similar observations for Clevrtex and Movi-C. The value channel has little to no impact, which is expected as it highly correlates with the RGB channels Table 7. The main reason for the worse segmentation quality of the RGB baseline is that models degenerated for some seeds (2 for 4, 5 for 24). The FG-ARI score for the best-performing seed of RGB models is 84.1 for the 4 variant and 76.6 for the 24, competing with the enhanced color spaces. Degenerated models also lead to a higher MSE. Contrarily, only one model of the RGB-S color space degenerated for the 24 version.

### 5.1.3 CLEVRTEX

Clevrtex (Karazija et al., 2021), conceptually similar to Clevr but having more shapes, sizes, and, most importantly, a manifold of photo-realistic textures (see Figure 3), is the most challenging dataset we considered. We use the invariant SA architecture by Biza et al. (2023) and similarly show two variants of model complexities: We use a standard five-level CNN autoencoder (Figure 3 for details), labeled CNN in the following, and also a Resnet34 architecture which promised significant performance gains (labeled ResNet). The RGB baselines can mostly segment scenes, but small objects are often encoded with other slots or multiple slots encode one object. Adding the saturation (and value) channels significantly improves the segmentation quality - The composite color spaces improve the baselines for 10% in absolute FG-ARI score, independent of the model's complexity. The CNN models on the enhanced color spaces achieve segmentation results comparable to the ResNet model trained on RGB while the encoder has approximately 20 times fewer parameters. Surprisingly, models trained on the HSV color space work best for both model architectures. We conclude that the hue channel, although unsteady, can be a valuable objective for more complex scenes - it is uncorrelated to the other dimensions, and the objects are not determined by a *single* color but can be composed of multiple colors - the discontinuity in the "Hue" channel might thus not be as influential. Further research on more complex datasets is necessary to confirm out hypothesis. We also evaluated the models *trained on the standard Clevrtex dataset* on an out-of-distribution test set containing novel shapes and materials. All models are robust and maintain almost the same segmentation quality (see subsection D.1). The reconstruction error increases significantly for all model variants.

### 5.1.4 MOVI-C

Originally, Movi-C (Greff et al., 2022) is a collection of short video clips. For this work, we consider its snapshots as an image dataset, similar to Seitzer et al. (2023); Fan et al. (2024). Investigating the effect of combined color spaces for videos is an interesting future work. Movi-C consists of 3 to 10

photorealistic objects on a background and is yet another significant step up in complexity. While SA trained on RGB was previously reported to fail on it (Seitzer et al., 2023), self-supervised signals provided a significant performance step-up. Our experiments utilize the same ResNet architecture as used for Clevrtex. Our object discovery measures for RGB are consistent with the related literature (Seitzer et al., 2023) - although the FG-ARI scores are similar across color spaces, we observed a qualitative difference in segmentation: The low segmentation scores for RGB originate from slots that segment scenes into areas. Contrarily, RGB-S and RGB-HSV attend to objects but often split the object into multiple slots (Figure 3). This observation is quantitatively captured by the mIoU score, which is significantly higher for the enhanced color spaces. While the enhanced color spaces improve object discovery, they do not suffice to match the performance of self-supervised signals for Movi-C (Seitzer et al., 2023) (see subsection D.6). However, we also note that the RGB-HSV models provide the best mIoU scores but lack proper reconstruction quality. More powerful decoders (in combination with additional color channels) might thus help to enhance the segmentation performance further.

## 5.2 RESULTS ON UNDERLYING GENERATING FACTORS

We tested how well the slots represent the objects they binded. We, therefore, run two tests to predict their underlying generating factors (Locatello et al., 2019). The first test is to train a linear and shallow non-linear predictor that should map from the slot representations to the factors of variation of the Clevrtex objects: Those are uniformly sampled from four shapes, three sizes, and 59 materials. We matched objects to slots based on the maximal mask overlap to construct the dataset to train and evaluate the predictor. Average precision is shown in Figure 4 (left and middle). Although the quality of slot representations was shown to correlate with the FG-ARI (Wenzel et al., 2022), the quality of slot representations differs strongly, especially for the materials: The HSV and RGB-S model at least double the precision of the RGB model. While the additional color dimensions do not directly influence the size and shape, the material is mainly determined by hue, saturation, and value. We suspect that the explicit representation helps slots generalize to underlying factors.

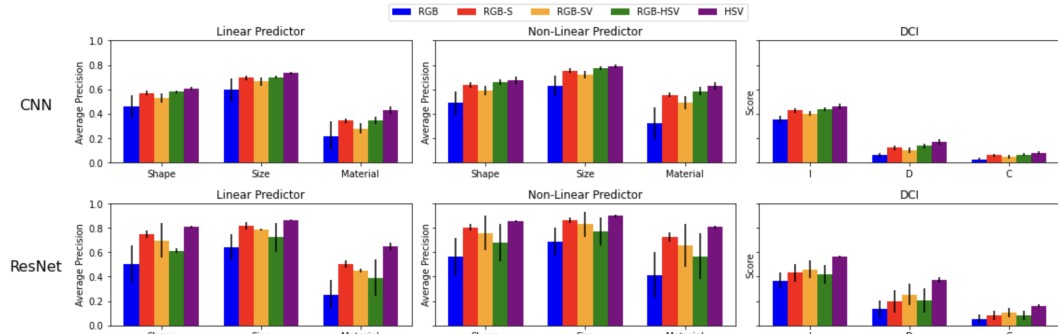

Figure 4: We report Slot Learning and Disentanglement results for the CNN and ResNet Variants of the Clevrtex dataset. The left figure shows the average precision of a linear predictor trained to map from slot representations to underlying object properties. The middle figure shows results for a shallow non-linear predictor. The right figure shows the Informativeness, Disentanglement, and Completeness of slot representations (Eastwood & Williams, 2018). The composite color spaces consistently outperform the RGB space for all considered metrics. The HSV space even outperforms the composite color spaces. Although the composite color spaces significantly improve the representative power of underlying object factors, they still show low performance in disentanglement and completeness.

Secondly, we employed the DCI metrics (Eastwood & Williams, 2018; Li et al., 2020) (see subsection A.3) to assess slot representations further. We follow a similar procedure as Singh et al. (2023) but employ a decision tree classifier instead of gradient-boosted trees for efficiency. Similar to our first test, all models improve over the ones trained on RGB. However, the plain HSV models, trained on the fully decorrelated color space, perform significantly better than all other models. Similarly, all ResNet models improve over their CNN counterpart. Although Seitzer et al. (2023) have shown

that scale alone is not enough to learn object representations, it seems that the simple CNN lacks the complexity to generalize to underlying factors. Overall, all models show low performance in disentanglement and completeness. An interesting future work would be to combine the color spaces with models explicitly disentangling factor representations (Singh et al., 2023; Stammer et al., 2024). We extend our analysis on slot representations in subsection D.4 and subsection D.5. We observed that slots obtained from composite targets tend to be more robust against lightning changes, and slots obtained from RGB targets tend to learn spatial areas instead of object representations.

## 6 LIMITATIONS AND FUTURE WORK

By only requiring a different output representation, our approach is generally applicable to all visual scenes; nonetheless, there are several limitations in our approach and evaluation: We only considered preexisting color channels of the *most common color spaces*, and there may be better composite channels suited for object detection. Analyzing whether a non-correlated RGB space, e.g., by applying PCA, already improves the scene segmentation is a promising starting point. Furthermore, we base our argument on the non-correlation to detect suitable complementary color channels. However, *more elaborate techniques* considering the data distribution of pixels may be needed to filter for complementary dimensions efficiently. Furthermore, we only evaluated our enhanced color spaces for *OCRL tasks*, but they might generally apply to a broader field of computer vision. At the same time, we based our discussion only on *natural images*. Finally, we only tested and compared the enhanced color spaces on SA - however, recent approaches based on self-supervised learning and pre-trained encoders have shown promising results - using composite color spaces (e.g., in the pretraining method) might lead to further performance improvement, but is impossible to assess a priori. An interesting direction for future research is if the performance leaps are achieved by using composite color space transfer to other unsupervised networks for visual computing. We started the elaboration of this paper by investigating the influence of the visual scene data representation on the performance of unsupervised OCRL. Our work is a *starting point*, showing promising performance without negative effects in OCRL, bearing the potential for numerous research directions: A first step is to evaluate the *design space of combined color spaces* - other channels than saturation may bear similar potentials, providing a consistent target to discriminate between objects. Furthermore, we designed the color space for OCRL, but combining color spaces got less attention in *computer vision in general*. However, a combination of color spaces still needs to be explored.

## 7 CONCLUSION

We challenge the common assumption that RGB images are the optimal color space for unsupervised learning in computer vision (Caron et al., 2021; Radford et al., 2021; Locatello et al., 2020) and demonstrate this for OCRL. We highlighted that the RGB space is not only highly correlated, effectively reducing the learned target to one dimension, but that composite color spaces offer non-correlated dimensions while upholding expressive features for OCRL. In a evaluation, we tested the LAB and the HSV color space as complimentary targets and found that HSV shows interesting properties due to its non-correlatedness and similarity to human perception. However, while HSV improves models on some datasets, it leads to degeneration for other datasets due to the discontinuity in the Hue channel. We thus combined the stability of the RGB space with the uncorrelated, expressive channels of HSV, constructing combined color spaces. Training models on the combined color spaces leaves the main *architecture unchanged*, is *broadly applicable*, virtually *cost-free*, and significantly *improves performance* on object detection. Especially the *RGB-S* space, combining the RGB space with the Saturation channel (significantly) enhances performance on object discovery and scene segmentation across all five datasets. Most significantly, on the challenging *Clevrtex* dataset, we increase the *FG-ARI* from 75.6 to 92.7. We further show evidence that the combined color spaces also improve performance on photo-realistic datasets such as Movi-C and improve the *mIoU* from 21.3 to 27.2. Similarly, we highlight that training on the pure HSV color space can sometimes improve the FG-ARI further but lacks stability on other datasets. Moreover, we evaluated the extracted slots' capability to bind object property and noticed significant improvements from 27.1 % to 57.8 % average precision. To summarize, the composite RGB-S and RGB-SV color space (significantly) *outperforms RGB in any experiment* without negative side effects or additional cost.

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

## A    REPRODUCIBILITY OF EXPERIMENTS

Our work contains a wide range of empirical evaluations. The following section serves as documentation to reproduce our experiments. We provide model architectures, optimization hyperparameters, and further details.

### A.1    MODEL ARCHITECTURE

We consider a wide range of complexities in our datasets. While Clevr contains only simple geometric objects on a gray background, Movi-C depicts photorealistic textures and backgrounds. We adjust the scale of model architectures accordingly. All models consist of an encoder, the Slot Attention module, and the spatial broadcast decoder. We report details of the Table A.1.1, Table A.1.2, Table A.1.3. For the Clevr dataset, we apply vanilla Slot Attention, for all other datasets we apply position and scale invariant slot attention (Biza et al., 2023).

#### A.1.1    ENCODER DETAILS

Table 3: Architecture Details of CNN encoder.

|  | Layer | Kernel Dimension | Kernel Size | Stride | Padding | Activation |
|---|---|---|---|---|---|---|
| *Clevr* | 1 | 64 | 5x5 | 1x1 | Same | ReLu |
|  | 2 | 64 | 5x5 | 1x1 | Same | ReLu |
|  | 3 | 64 | 5x5 | 1x1 | Same | ReLu |
|  | 4 | 64 | 5x5 | 1x1 | Same | ReLu |
| *Multishapenet* | 1 | 64 | 5x5 | 2x2 | Same | ReLu |
|  | 2 | 64 | 5x5 | 2x2 | Same | ReLu |
|  | 3 | 64 | 5x5 | 2x2 | Same | ReLu |
|  | 4 | 64 | 5x5 | 1x1 | Same | ReLu |
| *Clevrtex* | 1 | 64 | 5x5 | 2x2 | Same | ReLu |
|  | 2 | 64 | 5x5 | 2x2 | Same | ReLu |
|  | 3 | 64 | 5x5 | 2x2 | Same | ReLu |
|  | 4 | 64 | 5x5 | 1x1 | Same | ReLu |

#### A.1.2    SLOT ATTENTION MODULE DETAILS

We use Slot Attention with three iterations and a residual connection as proposed by Locatello et al. (2020). For all datasets except Clevr, we utilize the position-invariant variant of Biza et al. (2023). The number of slots is set to the maximum number of objects in the dataset plus the background, resulting in eleven slots for all datasets except Multishapenet, where we use five slots. The hyperparameters of the Slot attention module are depicted in Table A.1.2.

#### A.1.3    DECODER DETAILS

The hyperparameters of the spatial decoder are detailed in Table A.1.3.

### A.2    OPTIMIZATION PARAMETERS

Similar to Locatello et al. (2020); Biza et al. (2023), we apply the Adam optimizer. We use a batch size of 32 with a learning rate of $2 \cdot 10^{-4}$ for $250k$ steps. We use a learning rate warm-up from 0 to $50k$ steps, and afterwards a cosine learning rate decay for $100k$ steps. Due to the large variance for object segmentation, we train all CNN models on 10 random seeds, and all ResNet models on 6 random seeds.

Table 4: Architecture Details of Slot Attention Module

|  | Name | Dimension | Activation |
|---|---|---|---|
| *Clevr* | Position Encoding | 64 | ReLu |
|  | Key | 64 | None |
|  | Query | 64 | None |
|  | Value | 64 | None |
|  | GRU | 64 | ReLu |
|  | Residual MLP | $128 \rightarrow 64$ | ReLu $\rightarrow$ Relu |
| *Multishapenet* | Position Encoding | 64 | ReLu |
|  | Key | 64 | None |
|  | Query | 64 | None |
|  | Value | 64 | None |
|  | GRU | 64 | ReLu |
|  | Residual MLP | $128 \rightarrow 64$ | ReLu $\rightarrow$ Relu |
| *Clevrtex CNN* | Position Encoding | 64 | ReLu |
|  | Key | 64 | None |
|  | Query | 64 | None |
|  | Value | 64 | None |
|  | GRU | 64 | ReLu |
|  | Residual MLP | $128 \rightarrow 64$ | ReLu $\rightarrow$ Relu |
| *Clevrtex Resnet* | Position Encoding | $256 \rightarrow 128$ | ReLu |
|  | Key | 128 | None |
|  | Query | 128 | None |
|  | Value | 128 | None |
|  | GRU | 128 | ReLu |
|  | Residual MLP | $256 \rightarrow 128$ | ReLu $\rightarrow$ Relu |
| *Movi-C* | Position Encoding | $256 \rightarrow 128$ | ReLu |
|  | Key | 128 | None |
|  | Query | 128 | None |
|  | Value | 128 | None |
|  | GRU | 128 | ReLu |
|  | Residual MLP | $256 \rightarrow 128$ | ReLu $\rightarrow$ Relu |

## A.3 EVALUATION METRICS

While the metrics for object discovery are well established for OCRL (Locatello et al., 2020; Biza et al., 2023; Seitzer et al., 2023), we provide more details about our experiments of underlying generating factors. There we use the DCI metrics of Eastwood & Williams (2018). According to Eastwood & Williams (2018), those metrics measure:

- **Disentanglement**: "The degree to which a representation factorizes or disentangles the underlying factors of variation, with each variable (or dimension) capturing at most one generative factor"

- **Completeness**: "The degree to which a single code variable captures each underlying factor"

- **Informativeness**: "The amount of information that a representation captures about the underlying factors of variation"

The classifier and DCI metrics require slots to be matched to a ground truth object. We match pairs by the largest overlap of the ground truth mask with the decoded alpha mask of slots. For the property prediction experiments, we trained two classifiers. The linear one directly maps from the slot representations to one-hot encoded property predictions, while the non-linear one additionally includes a ReLu-activated intermediate layer with 128 dimensions. We utilize an external library available at Loc for the DCI experiments. We use a decision tree classifier with a batch size of 32.

Table 5: Architecture Details of CNN decoder

| | Layer | Kernel Dimension | Kernel Size | Stride | Padding | Activation |
|---|---|---|---|---|---|---|
| *Clevr* | 1 | 64 | 5x5 | 1x1 | Same | ReLu |
| | 2 | 64 | 5x5 | 1x1 | Same | ReLu |
| | 3 | 64 | 5x5 | 1x1 | Same | ReLu |
| | 4 | 64 | 5x5 | 1x1 | Same | ReLu |
| | 5 | 64 | 5x5 | 1x1 | Same | ReLu |
| | 6 | Color Channels + 1 | 3x3 | 1x1 | Same | None |
| *Multishapenet* | 1 | 64 | 5x5 | 2x2 | Same | ReLu |
| | 2 | 64 | 5x5 | 2x2 | Same | ReLu |
| | 3 | 64 | 5x5 | 2x3 | Same | ReLu |
| | 4 | 64 | 5x5 | 1x1 | Same | ReLu |
| | 5 | 64 | 5x5 | 1x1 | Same | ReLu |
| | 6 | Color Channels + 1 | 3x3 | 1x1 | Same | None |
| *Clevrtex CNN* | 1 | 64 | 5x5 | 2x2 | Same | ReLu |
| | 2 | 64 | 5x5 | 2x2 | Same | ReLu |
| | 3 | 64 | 5x5 | 2x3 | Same | ReLu |
| | 4 | 64 | 5x5 | 1x1 | Same | ReLu |
| | 5 | 64 | 5x5 | 1x1 | Same | ReLu |
| | 6 | Color Channels + 1 | 3x3 | 1x1 | Same | None |
| *Clevrtex Resnet* | 1 | 128 | 5x5 | 2x2 | Same | ReLu |
| | 2 | 128 | 5x5 | 2x2 | Same | ReLu |
| | 3 | 128 | 5x5 | 2x3 | Same | ReLu |
| | 4 | 64 | 5x5 | 1x1 | Same | ReLu |
| | 5 | 64 | 5x5 | 1x1 | Same | ReLu |
| | 6 | Color Channels + 1 | 3x3 | 1x1 | Same | None |
| *Movi-C* | 1 | 128 | 5x5 | 2x2 | Same | ReLu |
| | 2 | 128 | 5x5 | 2x2 | Same | ReLu |
| | 3 | 128 | 5x5 | 2x3 | Same | ReLu |
| | 4 | 64 | 5x5 | 1x1 | Same | ReLu |
| | 5 | 64 | 5x5 | 1x1 | Same | ReLu |
| | 6 | Color Channels + 1 | 3x3 | 1x1 | Same | None |

### A.4 IMPLEMENTATION OF COLOR SPACES

In the main text, we regularly discuss RGB, CIELAB, and HSV as color spaces. However, while the term "color spaces" is regularly used, it is imprecise: For example, RGB comprises a collection of *additive* color spaces, and arguably its most important representative is the standard RGB (sRGB) space - images in the web are usually represented by the sRGB space if not tagged otherwise. For our paper, we do not discriminate between the different implementations, and our reasoning is not based on a specific implementation. Contrarily, we argue that adding additional color channels leads to output representations that are less statistics-based but lead to more semantic segmentations. We utilize the Tensorflow implementation (tfi) to convert images to other color spaces.

## B DATASETS

### B.1 CLEVR

Clevr (Johnson et al., 2016) is arguably the least complex dataset we consider. Three to ten simple geometric objects are randomly placed on a gray background. The number of objects is uniformly sampled, containing four factors of variations: Objects differ between three shapes, two sizes, two materials, and eight colors. The camera and light source are randomly jittered. We trained our models on the whole train set, consisting of 70000 images, and evaluated on 5000 images randomly sampled from the test set. We preprocess the images by taking a 192x192 center crop of the images and then bilinearly rescale them to 128x128. We do the same for the ground truth masks but rescale them with nearest-neighbor interpolation. The dataset with ground truth masks is available at kub.

## B.2 MULTISHAPENET

Multishapenet (Stelzner et al., 2021) is a significant step up in complexity compared to Clevr. Although the background is also gray, the objects are visually more complex: Each of the objects falls into one of the categories of chairs, tables, and cabinets (uniformly sampled), which are then used to sample photorealistic models from the ShapeNetV2 set (Chang et al., 2015). Multishapenet contains 11733 unique shapes. Camera and light are randomly jittered, as in Clevr. We preprocess the images by taking a 192x192 center crop of the images and then bilinearly rescale them to 128x128. We do the same for the ground truth masks but rescale them with nearest-neighbor interpolation. The training set contains 70000 images, while the test set contains 15000. We evaluate 5000 of them. We filter for images containing exactly four objects for the Multishapenet 4 version. The dataset is available at (msn).

## B.3 CLEVRTEX

Clevrtex (Karazija et al., 2021) is conceptually similar to Clevr. Each scene includes 3 to 10 random objects. However, it contains 60 photorealistic backgrounds with complex textures. Similarly, the objects are sampled from four shapes, three sizes, and 60 complex textures. Scenes are often dark due to the backgrounds and objects. The camera is chosen as in Clevr, but Clevrtex includes complex lighting effects through three light sources. We apply the same cropping and resizing strategy as for Clevr. Clevrtex includes 40000 images for training and 5000 images for training. We also evaluate our models on an out-of-distribution set containing four unseen shapes and 25 additional textures. Clevrtex (v2) can be downloaded at cle.

## B.4 MOVI-C

Movi-C is a video dataset consisting of 24 frames showing objects falling to the ground with realistic physics. Each scene consists of 3 to 10 objects with a complex background. Contrarily to Clevrtex, the objects are not simple geometric shapes but are sampled from 11 realistic categories, such as "Toys" or "Car Seat." The lighting effects originate from the sampled background and thus differ strongly in their lightness. Similar to related works (Seitzer et al., 2023), we treat the dataset as an image dataset, taking all 24 snapshots. That yields 234000 images for training and 6000 for testing. We ensure no overlap between the train and test set; all images originate from different videos. Movi-C can be downloaded at mov.

# C COLOR SPACES AND THE BIOLOGICAL VISUAL SYSTEM

We predominantly used the RGB and HSV color space in the main text for our experimental evaluation. We motivate their usage in subsection 3.2 with analogies to the human visual system. This section serves as a basic introduction to the color perception of humans. We refer to Müller & (Lipsk) and Kim et al. (2009) for details.

Almost all theories of human color vision are based on the zone model of Müller & (Lipsk); Kim et al. (2009). Physical light waves stimulate three types of cones (short-, middle-, and long-wave) and rods in the pupil. Neurons combine those physical signals, yielding achromatic brightness, hue, and colorfulness sensations. Those are finally propagated with nerve fibers to the visual cortex. While the RGB color space is designed in analogy to the rudimentary physical activations of the cones, the HSV space is closely related to the sensations propagated to our visual cortex (Smith, 1978; Schwarz et al., 1987). In our work, we report that using additional color channels closely related to how humans perceive the world significantly improves the object detection capabilities of unsupervised OCRL networks.

# D EXTENDED EXPERIMENTAL RESULTS

## D.1 CLEVRTEX OUT OF DISTRIBUTION

The Clevrtex dataset includes an out-of-distribution dataset containing novel objects and textures different from those in the train set. We also evaluated the models on this test set for object discov-

ery. As models trained on the augmented color spaces better generalize to underlying generating factors, we hypothesized that they are more sensitive to out-of-distribution data with newly introduced factors of variation. However, all models perform similarly in terms of object discovery, as seen in Table D.1.

Table 6: Evaluation results on the out-of-distribution test set of Clevrtex for the different predicted output color spaces.

| | V | Pred. | FG-ARI ↑ | mIoU ↑ | MSE* ↓ |
|---|---|---|---|---|---|
| | | RGB | 65.6 ± 16.0 | 39.5 ± 12.1 | **429.7 ± 49.5** |
| | | RGB-S | **82.5 ± 3.2** | 57.8 ± 3.6 | 521.4 ± 17.4 |
| | CNN | RGB-SV | 79.9 ± 3.5 | 50.9 ± 7.3 | 488.6 ± 33.8 |
| | | RGB-HSV | **83.3 ± 1.9** | 59.8 ± 4.2 | 533.9 ± 22.8 |
| Clevrtex OOD | | **HSV** | **83.6 ± 1.5** | **64.9 ± 2.0** | 591.5 ± 19.4 |
| | | RGB | 74.6 ± 11.5 | 45.3 ± 14.2 | **391.2 ± 66.6** |
| | | RGB-S | 90.4 ± 2.8 | 69.4 ± 3.7 | 509.1 ± 37.6 |
| | ResNet | RGB-SV | 87.3 ± 10.2 | 65.9 ± 14.9 | 480.2 ± 77.9 |
| | | RGB-HSV | 83.1 ± 11.4 | 56.7 ± 14.8 | 446.1 ± 81.2 |
| | | **HSV** | **92.6 ± 0.8** | **77.3 ± 1.6** | 636.1 ± 19.7 |

*\* MSE caveat: A tendency for higher values for the enriched color spaces and HSV is expected as the MSE involves a conversion to RGB.*

## D.2 CORRELATIONS OF COLOR CHANNELS

In the main text, we argued that the RGB channels are highly correlated in natural images. We empirically verified that by measuring the Pearson correlation coefficient for three well-known datasets with natural lighting, namely *ClevrTex* (Karazija et al., 2021), *MS Coco* (Lin et al., 2014), and *ImageNet* (ILSVRC2012). We sampled 1000 images per dataset and measured the pixel-wise correlation between the color channels. We report the results in Table 7. Additionally, we also measured the correlation for the HSV channel. All channels of RGB strongly correlate with the value that captures the lightness (see subsection D.8). The hue and saturation channels are mostly non-correlated. We show in the main text that those channels provide complementary information useful for unsupervised OCRL.

Table 7: Correlation of Color channel pixel values for the three datasets *ClevrTex* (Karazija et al., 2021), *MS Coco* (Lin et al., 2014), and *ImageNet* (ILSVRC2012) (Deng et al., 2009).

| | Channel | R | G | B | H | S | V |
|---|---|---|---|---|---|---|---|
| | R | · | +.91 | +.78 | −.16 | −.19 | +.98 |
| | G | | · | +.94 | ±.00 | −.44 | +.94 |
| ClevrTex | B | | | · | +.17 | +.63 | +.82 |
| | H | | | | · | −.33 | −.08 |
| | S | | | | | · | −.19 |
| | R | · | +.91 | +.77 | −.12 | −.36 | +.94 |
| | G | | · | +.91 | −.01 | −.47 | +.94 |
| MS Coco | B | | | · | +.19 | −.56 | +.87 |
| | H | | | | · | −.05 | +.01 |
| | S | | | | | · | −.28 |
| | R | · | +.89 | +.77 | −.24 | −.57 | +.85 |
| | G | | · | +.89 | −.12 | −.43 | +.94 |
| ImageNet | B | | | · | +.10 | −.34 | +.93 |
| | H | | | | · | +.20 | −.05 |
| | S | | | | | · | −.25 |

## D.3    INPUT TRANSFORMATIONS

In our main text, we mainly argued for color space transformations in the *targets*. Color space transformations in the inputs were already investigated in the literature (see section 4) - depending on the task and data, other color spaces than RGB have shown performance leaps. However, we argued that any machine learning model can closely approximate the color space transformations, which makes color space transformations in the inputs less convincing for OCRL. The situation is different for transformations in the output, as the slots must represent those features. To test our hypothesis, we run experiments with transformations in the input and output. Therefore, we used the CNN Clevrtex model on three random seeds each. We report the FG-ARI in Table 8.

Table 8: Object Discovery FG-ARI ($\uparrow$) for input/output transformations.

| Input/Output | RGB | HSV | LAB |
|---|---|---|---|
| RGB | $71.7 \pm 1.4$ | $86.7 \pm 1.7$ | $71.0 \pm 10.0$ |
| HSV | $68.3 \pm 7.0$ | $75.8 \pm 15.4$ | $74.7 \pm 15.6$ |
| LAB | $74.5 \pm 9.5$ | $86.2 \pm 1.5$ | $78.8 \pm 7.4$ |

## D.4    ROBUSTNESS UNDER LIGHTING CONDITIONS

We extend our experiments from the main text by testing how training on composite color spaces influences the robustness of models to distribution shifts. In the following, we explore how models behave when exposed to differing *lighting*. In section 3, we argued about the significant impact of lighting on the RGB channels. In contrast, most HSV channels are only slightly perturbed. We will explore whether those unperturbed channels transfer to more robust slot representations under lighting conditions.

We generate a novel test set similar to Clevrtex scenes for our experiments, utilizing the same codebase (Cle). However, for each generated scene, we generate three additional scenes where only the lighting conditions are varied. Specifically, we push the light source apart for 5, 10, and 20 units, respectively, resulting in gradually darker scenes. Qualitative samples for differing lighting conditions are shown in Figure 8. The effects on the color channels may be observed in Figure 5: While the R, G, B, and V channels are highly dependent on the lighting conditions, the hue and saturation show only moderate changes. We create a total of 1024 scenes.

Table 9: Object Discovery FG-ARI ($\uparrow$) and mIoU ($\uparrow$) for various color spaces (RGB2X) and differing lighting conditions.

| | Metric | Pred. | $L = 0$ | $L = 5$ | $L = 10$ | $L = 20$ |
|---|---|---|---|---|---|---|
| *Clevrtex Lighting* | FG-ARI $\uparrow$ | RGB | $73.9 \pm 7.4$ | $73.5 \pm 7.5$ | $70.9 \pm 6.9$ | $68.2 \pm 6.0$ |
| | | RGB-S | $85.9 \pm 4.3$ | $85.7 \pm 3.7$ | $83.0 \pm 3.3$ | $79.1 \pm 3.0$ |
| | | HSV | $87.9 \pm 1.3$ | $86.9 \pm 1.2$ | $84.6 \pm 0.8$ | $81.4 \pm 0.9$ |
| | mIoU $\uparrow$ | RGB | $44.4 \pm 9.6$ | $40.3 \pm 8.2$ | $35.8 \pm 6.5$ | $33.6 \pm 5.7$ |
| | | RGB-S | $62.6 \pm 3.7$ | $59.1 \pm 3.6$ | $52.8 \pm 3.6$ | $48.5 \pm 3.3$ |
| | | HSV | $72.1 \pm 1.5$ | $69.1 \pm 1.5$ | $62.8 \pm 2.1$ | $57.6 \pm 2.2$ |

We first test whether models trained on the RGB, RGB-S, and HSV space remain robust in discovering objects (see Table D.4). While the distribution shift decreases the segmentation of all models, independent of the target color space, they remain remarkably robust. Interestingly, even in the darkest scenes (distance $L = 20$), the RGB-S and HSV models still achieve a better performance than the RGB models without distribution shift. However, the RGB models are also only marginally influenced by darker scenes. We suspect that the Clevrtex training set (containing 40k images) contains enough scenes with dark materials to create models that are robust toward lighting effects inherently.

Secondly, we test whether the induced slot representations remain similar under different lighting conditions. Therefore, we first match slots to ground-truth objects based on their predicted mask.

Table 10: Slot representation distance for various color spaces (RGB2X) and differing lighting conditions. We report the Cosine and Euclidean distances for slots binding the same object under differing lighting conditions. "$0 \rightarrow L$" measures the (euclidean/cosine) distance between the slot obtained under normal lighting conditions and the lighting conditions observed as distance $L$.

| | Metric | Pred. | $0 \rightarrow 5$ | $0 \rightarrow 10$ | $0 \rightarrow 20$ |
|---|---|---|---|---|---|
| *Clevrtex Lighting* | Cosine $\rightarrow$ | RGB | $37.4 \pm 3.1$ | $59.8 \pm 5.7$ | $76.8 \pm 8.2$ |
| | | RGB-S | $29.8 \pm 2.2$ | $46.1 \pm 3.6$ | $59.2 \pm 4.1$ |
| | | HSV | $23.3 \pm 1.5$ | $35.2 \pm 2.8$ | $45.9 \pm 3.5$ |
| | Euclid. $\rightarrow$ | RGB | $58.1 \pm 3.0$ | $76.4 \pm 4.3$ | $87.8 \pm 5.3$ |
| | | RGB-S | $49.2 \pm 2.1$ | $64.7 \pm 3.5$ | $75.0 \pm 3.4$ |
| | | HSV | $42.4 \pm 2.2$ | $55.6 \pm 3.0$ | $65.0 \pm 3.3$ |

Then, we compare the slot representations of darkened scenes to the slot representation without the darkening effect. We compare them based on the Euclidean and Cosine distances. For comparison, we normalize each model's Euclidean and Cosine distance based on the respective average distance of all slots.

We report the results in Table D.4. While the lighting conditions strongly influence slot representation trained on the RGB space, the slot representation trained on RGB-S and HSV remains more robust. We attribute this to the complementary hue and saturation channels robust to lighting conditions. The underlying slot representations learn those invariances, transferring to more stable slot representations.

### D.5  SLOT ATTENTION MAPS

We extend our analysis of learned slot parameters from subsection 5.2 by inspecting the attention maps for each slot. To get an overview of the whole model, we aim to visualize not only single scenes but the whole Clevrtex test set. For each scene, we compute the $\alpha$ mask for every slot and derive the weighted (x,y)-means of the mask per slot. The mean (x,y)-values (per slot) are presented in Figure 6 for six models trained on RGB, RGB-S, and HSV.

While all slots, independent of the color space, occupy a spatial area, the distributions of the center points of the attention maps vary: For four models of the RGB space, the attention maps centers differ only slightly - those represent degenerated slots, binding fixed-size grids instead of objects. In our experiments, this behavior appears regularly for models trained on RGB and is also observed in the related literature(Seitzer et al., 2023; Locatello et al., 2020). This behavior does not happen as frequently with composite color spaces. We suppose that the complementary information of the additional color channels provides targets that pressure models to discriminate between objects and backgrounds - those might be similar in the RGB space due to their strong correlation with lighting effects but strongly differ in the uncorrelated HSV space. Those features (e.g., in color and saturation) must be represented by the underlying slot representations.

### D.6  COMPARISON TO SELF-SUPERVISED METHODS

Although self-supervised targets are not the main focus of our research, recent related work (Seitzer et al., 2023) has highlighted their remarkable performance on challenging real-world datasets. Thus, we compare our composite color spaces to the recent Dinosaur (Seitzer et al., 2023) architecture. We run experiments on the Clevrtex and Movi-C datasets using the same hyperparameter setting as described in (Seitzer et al., 2023) for Movi-C. However, we exchange the Vit-s8 backbone for a ResNet34 variant for comparability to our networks. We utilize training signals from a DINO (Caron et al., 2021) Vit-b16. The Dinosaur models with the ResNet variant show superior performance over the composite color spaces on Movi-C (FG-ARI $53.1 \pm 1.3$, mIoU $22.7 \pm 0.6$) but lack performance on Clevrtex (FG-ARI $22.1 \pm 10.7$, mIoU $11.2 \pm 4.8$). Related work Zhao et al. (2024) optimized a Dinosaur model for Clevrtex that does not degenerate, but its performance (FG-ARI $\approx 80.0$) is still far behind the composite color spaces.

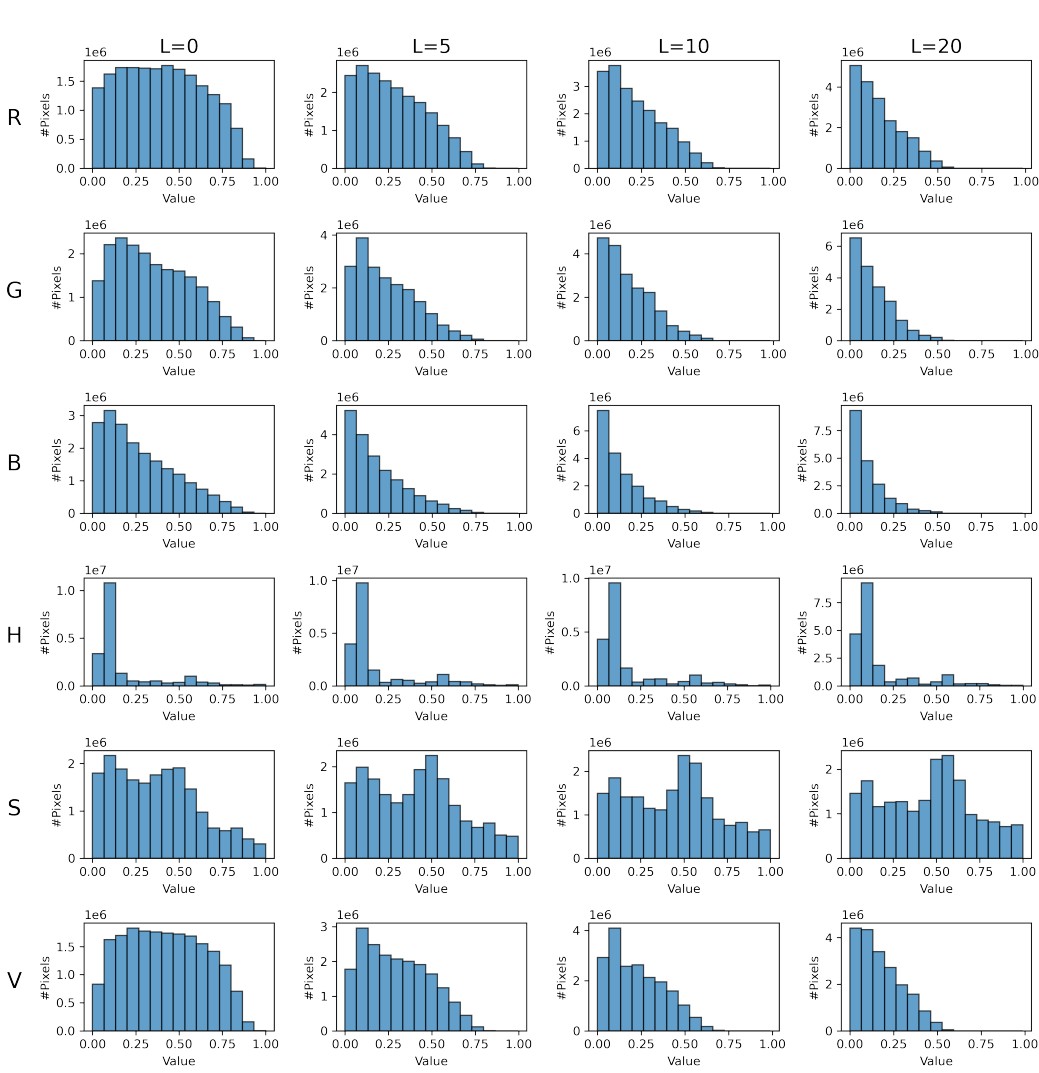

Figure 5: We show color statistics of our newly generated Clevrtex test set under different lighting conditions. The distances of the lights are denoted with the variable $L$, distancing all light sources. While the distribution of Hue and Saturation mostly remains consistent, all other color channels show immense discrepancies.

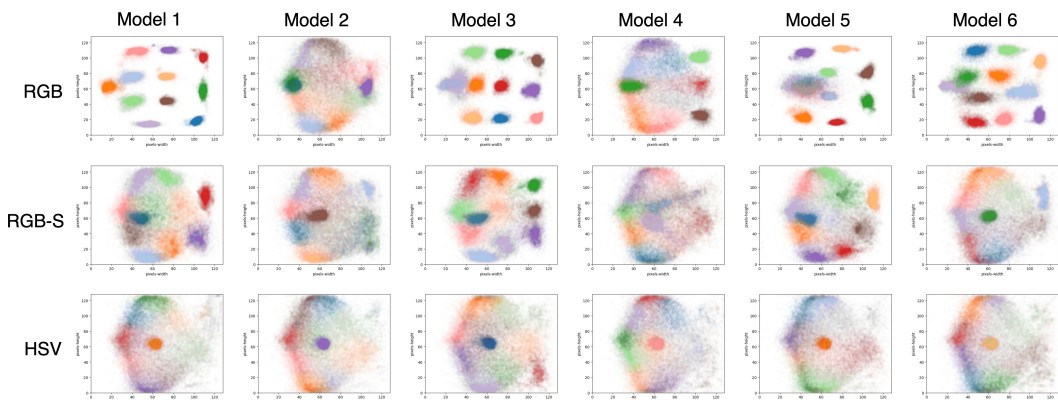

Figure 6: Detailed analysis for the Clevrtex test set: We plot the attention maps for models with a CNN backbone trained on different color spaces. Every dot in the scatter plot represents the spatial broadcast decoder's weighted mean $(x,y)$-$\alpha$ mask of a slot in a scene. The x- and y-axis represent the spatial layout of a Clevrtex scene with size $(128,128)$. Slots are discriminated by color, and we only plot slots that match an object based on the mIoU. While all slots bind to specific areas, most attention maps of slots in the HSV and RGB-S model show high variance, attending to objects close to that area. Most slots in the RGB model degenerate to fixed areas, not binding to specific objects. One can observe that the variance of the attention maps is higher (indicating a better binding) for HSV than RGB-S, in contrast to all other test sets.

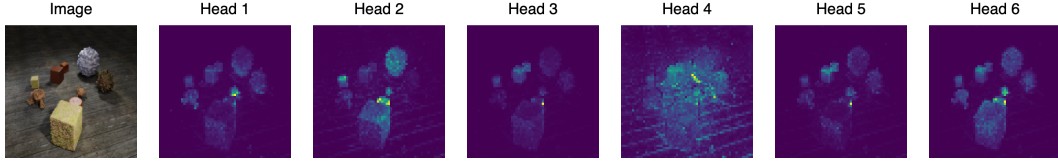

Figure 7: We visualize the self-attention maps of the [CLS] token of different heads of the last layer of a pre-trained DINO Vit-s8 (Caron et al., 2021) on a Clevrtex sample. For more information on the attention heads, see Din. Each of the attention heads focuses on a group of foreground objects, segmenting them from the background.

The extracted DINO features do not trivially segment the Clevrtex scenes into objects. Although the objects are segmented from the background (see the attention maps in Figure 7), the features do not naturally segment between objects. This can be observed by applying the trivial k-means clustering schemes on the DINO features. This strategy already outperforms SA networks trained on RGB losses in Seitzer et al. (2023) for the Movi and Coco datasets, but it shows poor performance on Clevrtex (see Table 11). Self-supervised signals might not be useful for those specific data distributions. Furthermore, while self-supervised signals show a massive improvement for real-world datasets, their performance is limited by the self-supervised signals. If information is not represented in these signals, the OCLR network can not recover these signals. Contrarily, composite color representations retain all information - thus, color representations might be used as complementary targets next to self-supervised signals or as sole targets with more elaborate networks.

Table 11: Object Discovery FG-ARI and mIoU for k-means clustering on DINO features

| Metric | Vit-s8 | Vit-s16 | Vit-b8 | Vit-b16 |
|---|---|---|---|---|
| FG-ARI (↑) | $50.1 \pm 13.3$ | $30.3 \pm 9.9$ | $21.3 \pm 7.1$ | $30.0 \pm 10.0$ |
| mIoU (↑) | $37.2 \pm 10.1$ | $19.4 \pm 5.9$ | $16.2 \pm 5.0$ | $19.6 \pm 5.8$ |

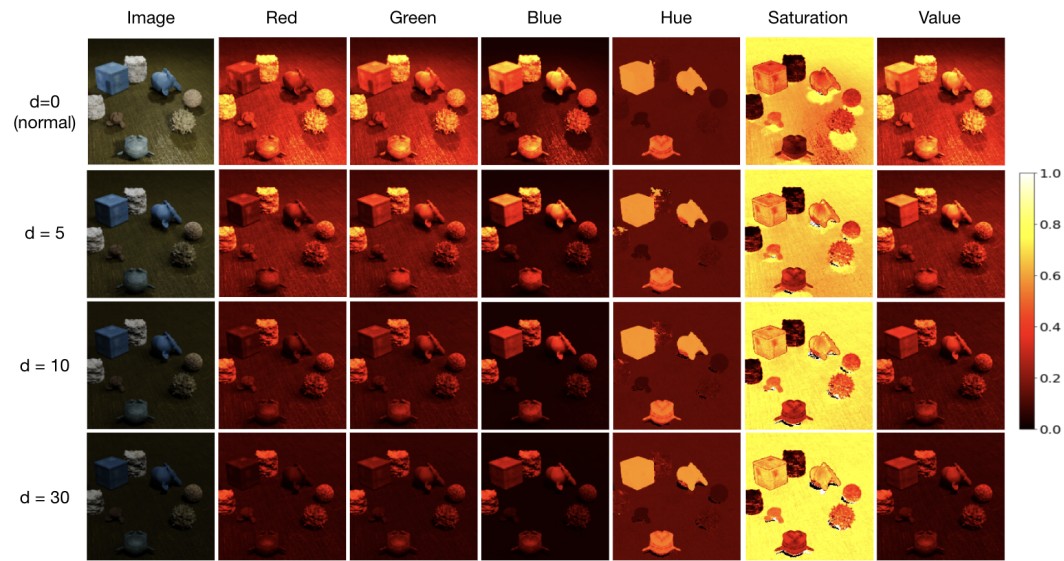

Figure 8: We show four scenes that are generated with the Clevrtex generator. All scenes contain the same objects and background; only the source of light is moved farther away. We plot the image together with heatmaps of the RGB and HSV channels.

## D.7 ABLATIONS ON THE NUMBER OF SLOTS

It was shown that the number of slots largely influences the performance of OCLR networks (Fan et al., 2024). Often, a lower number of slots than objects in the scene even helps to improve object detection. We thus run an ablation and test whether the composite color spaces improve OCLR networks. Table D.7 shows scene segmentation quality of CNN networks on the Clevrtex test set, trained with an ablated number of slots. We trained on three random seeds. Although the RGB models do not degenerate as often with a reduced number of slots, the RGB-S models outperform them consistently - the only exception is for a very low number of slots.

Table 12: Object Discovery FG-ARI and mIoU for an ablated number of slots on Clevrtex

|  | Metric | Pred. | 5 Slots | 7 Slots | 9 Slots |
|---|---|---|---|---|---|
| *Clevrtex* | FG-ARI ↑ | RGB | $65.1 \pm 5.1$ | $80.4 \pm 1.5$ | $74.3 \pm 11.1$ |
|  |  | RGB-S | $60.7 \pm 10.4$ | $83.1 \pm 4.4$ | $86.8 \pm 1.8$ |
|  | mIoU ↑ | RGB | $37.6 \pm 4.6$ | $50.0 \pm 2.2$ | $44.3 \pm 12.8$ |
|  |  | RGB-S | $22.2 \pm 12.1$ | $59.2 \pm 2.9$ | $65.4 \pm 0.7$ |

## D.8 QUALITATIVE LIGHTING EFFECTS ON CLEVRTEX

We qualitatively show how light effects influence color channels in Figure 8. Therefore, we created visual scenes with the same distribution as the Clevrtex dataset; however, we manipulated the light source, gradually moving it farther away. The objects are clearly discriminable in all channels at the start ($d = 0$). However, while the hue and saturation channels uphold discriminative features, all other channels lose them. Most objects blend into the background in the RGB and saturation channels for the darkest scene ($d = 0$). As the value of HSV captures the lightness, the hue, and saturation are less affected by light effects.

## D.9 QUALITATIVE SCENE SEGMENTATION

We show qualitative samples for all evaluation datasets in Figure 9, Figure 10, Figure 11, and Figure 12.

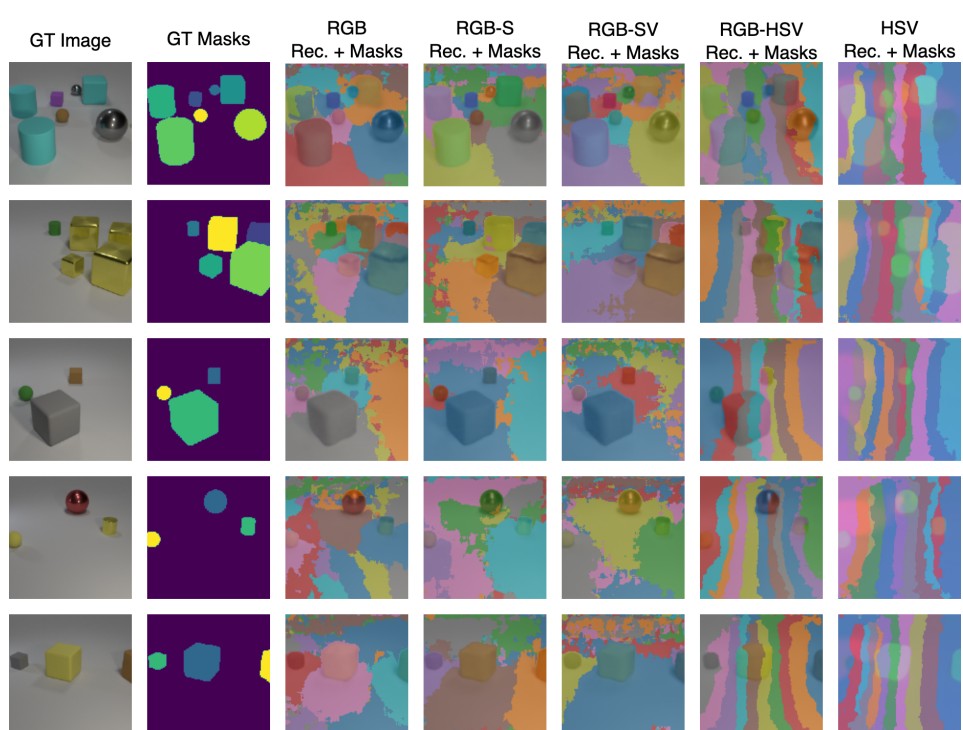

Figure 9: Exemplary scenes from the Clevr dataset.

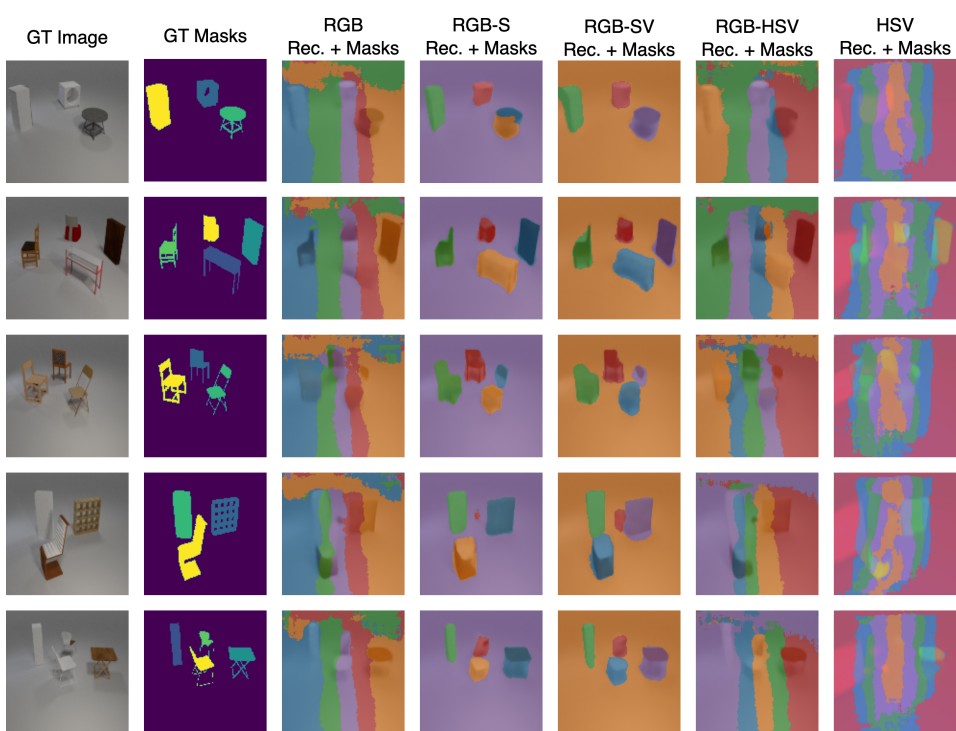

Figure 10: Exemplary scenes from the Multishapenet dataset.

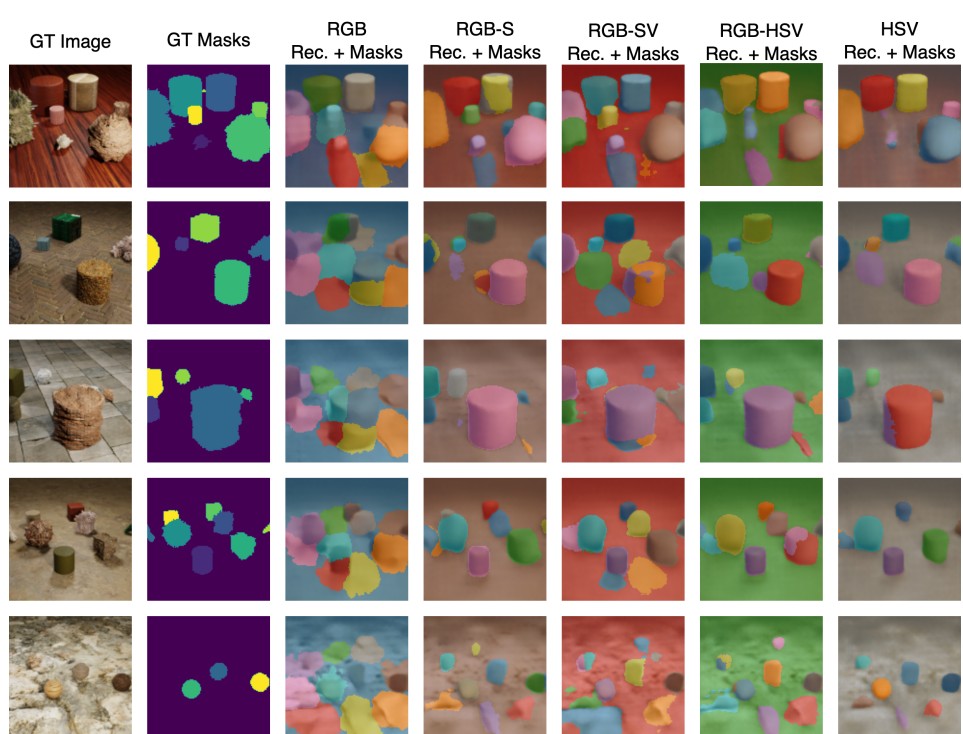

Figure 11: Exemplary scenes from the Clevrtex dataset.

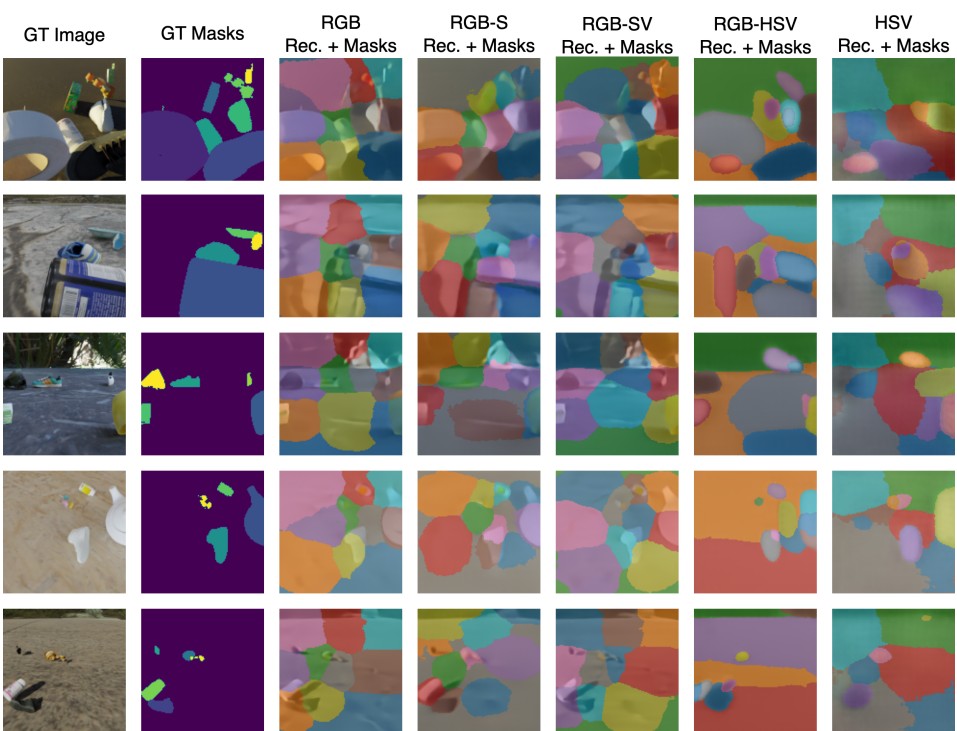

Figure 12: Exemplary scenes from the Movi-C dataset.

