# OpenReview forum: "Leveraging Color Channel Independence for Improved Unsupervised Object Detection"
_ICLR.cc/2025/Conference — ICLR 2025 Conference Withdrawn Submission_

### Official Review · Reviewer_aFR5 · 2024-11-03

**Soundness:** 2
**Presentation:** 3
**Contribution:** 2
**Rating:** 5
**Confidence:** 4

**Summary:**

The paper suggests that using RGB to encode input images is not the best input representation for object-centric representation learning. Instead, the paper shows that, in simple datasets, using RGB concatenated with HSV leads to better results.

**Strengths:**

I find very interesting the overall goal of the paper of enhancing the color representation.

**Weaknesses:**

Experiments are on synthetic datasets, or with simple and colorful objects. The selected datasets have few objects in front of simple backgrounds, making color information to be more useful than it would be in more challenging datasests with real images (e.g., COCO, ImageNet).

The experiments are very limited (as acknowledged in the conclusion). I praise the authors for acknowledging this, but I find that the limited experiments are insufficient to claim that building better color representations for the input would lead to improved performance in realistic image datasets beyond clever, …

I think the idea is good, but the authors should have tested the idea on more challenging datasets. It is likely that the simple approach proposed in the paper to extend color information would be insufficient when the dataset is more complex.

**Questions:**

1. would it be possible to test on more challenging datasets? Do you still expect a benefit there?

2. There are many color spaces that provide decorrelated channels. A simplest one is to use a linear rotation of the space spanned by the channels R,G and B by doing PCA on it.

---

> ### Author Response · Authors · 2024-11-27
>
> We thank the reviewer for the valuable feedback and hope we can address the questions below.
>
> - "Would it be possible to test on more challenging datasets? Do you still expect a benefit there?"
>
>     We do not expect a large benefit from more challenging datasets, as the models already struggle with the complexity of Movi-C. However, we argue that our experiments are on the same level of complexity as related works [1,2] that do not rely on additional data (such as self-supervised signals). We argue in Appendix, Section D.6, that research on color channels remains important, even though not yet applicable to real-world datasets.
>
> - "There are many color spaces that provide decorrelated channels. A simplest one is to use a linear rotation of the space spanned by the channels R,G and B by doing PCA on it."
>
>     As the design space of composite color spaces is combinatorial, we could not include an extensive evaluation of all color spaces in our analysis. However, we note that the decorrelation of color channels alone is insufficient to improve OCRL - for example, the LAB space is not correlated but does not improve over RGB representations. We agree that using a PCA-decorrelated RGB space is an interesting opportunity for future research, and we added it to our future work section.
>
> [1] O. Biza, S. V. Steenkiste, M. S. M. Sajjadi, G. F. Elsayed, A. Mahendran, and T. Kipf. 2023. Invariant slot attention: object discovery with slot-centric reference frames. In Proceedings of the 40th International Conference on Machine Learning (ICML'23), Vol. 202. JMLR.org, Article 106, 2507–2527.
>
> [2] G. Singh, Y. Kim and S. Ahn. 2022. Neural Systematic Binder. International Conference on Learning Representations.

---

> > ### Author Response · Authors · 2024-12-04
> >
> > We thank the reviewer for their constructive feedback and for highlighting the importance of evaluating our approach to more challenging datasets. We agree that extending our method to such datasets is essential for understanding its broader applicability and agree it is not straightforward. To address this, we have included a new section, Appendix D.6, in the uploaded new pdf manuscript. In this section, we provide an initial exploration of the requirements for applying our approach to more complex datasets and investigate more advanced feature extraction methods, including Dino and Dinosaur. We argued for conjunction with our proposed color space extensions (e.g., RGB-S and other combinations). As anticipated by the reviewer, directly applying our RGB-S and other variants with current object-centric frameworks, such as Slot Attention, does not scale effectively to more challenging datasets. We attribute this to inherent limitations of the Slot Attention mechanism and how it is currently implemented.
> >
> > To address these limitations, we investigated more advanced feature extraction methods, including Dino and Dinosaur, in conjunction with our proposed color space extensions (e.g., RGB-S and other combinations). Our results, which we detail in Appendix D.6 and also shared in intermediate feature map visualizations (in response to reviewer puBu, see comments and explanation above, https://github.com/ocrlAnonym/iclr_2025_slot), demonstrate improved spatial separation in intermediate feature maps. This improvement directly enhances the learned binding mechanisms within Slot Attention.
> >
> > Additionally, we observed that our method improves robustness to challenging factors such as lighting variations, where current models for large datasets do struggle, suggesting that the proposed approach has the potential to generalize to more realistic and complex datasets. While these results are preliminary, they highlight a promising direction for future work and underline the potential for our approach to contribute meaningfully to object-centric representation learning in real-world scenarios and more challenging scenarios.

---

### Official Review · Reviewer_puBu · 2024-11-04

**Soundness:** 3
**Presentation:** 3
**Contribution:** 3
**Rating:** 5
**Confidence:** 4

**Summary:**

The paper introduces a novel perspective on output target color space for object-centric representation learning. The authors provide an in-depth discussion of various color spaces as prediction targets, including RGB, HSV, and CIE-LAB. In particular, using Slot Attention as the backbone, the authors conduct experiments with different target color spaces, encompassing both individual and composite color spaces.

**Strengths:**

1) As most previous works focused on the different color space input, the exploration and discussion of using different color space as the target is beneficial for the community.
2) This paper finds that predicting composite color spaces offers a slight improvement over traditional RGB targets in unsupervised object-centric representation learning (OCRL). This approach not only inspires advancements in OCRL but also provides a direction for image modeling-based unsupervised representation learning.
3) The discussion of different color spaces is also beneficial for the computer vision community.

**Weaknesses:**

**1) Although the authors present an interesting approach to enhancing model understanding of foreground objects in a scene by predicting different color spaces, the paper primarily discusses the impact of various color spaces from a results-oriented perspective without providing a more in-depth analysis. For example, influence on model parameters and feature characteristics are lacking. I hope the authors can offer further insights into how different target color spaces influence the model.**

**2) Although the authors state that their work focuses on color spaces, the essence of their approach lies in enhancing the model's perceptual abilities by adjusting target objectives. Therefore, using only an RGB-RGB baseline may not be reliable. It would be more robust to include comparisons with methods that use different representations as targets, such as (Seitzer et al., 2023).**

**3) While the authors' idea is interesting, they do not provide a reliable analysis from a representation perspective on the inclusion of different color space targets. For instance, in Figure 2 of the main paper, there is a noticeable similarity between saturation and the mask in highlighting foreground objects; however, the mask is discrete and contains categorical information, while saturation focuses solely on object shape. There is insufficient technical analysis explaining the impact of using saturation.**

M. Seitzer, M. Horn, A. Zadaianchuk, D. Zietlow, T. Xiao, C.-J. Simon-Gabriel, T. He, Z. Zhang, B. Sch¨olkopf, T. Brox, and F. Locatello. Bridging the Gap to Real-World Object-Centric Learning, March 2023. URL http://arxiv.org/abs/2209.14860. arXiv:2209.14860 [cs].

**Questions:**

1) The authors only discuss grayscale for the single-channel approach. Considering the high correlation between RGB channels, what would the impact for OCRL be if using just one of the RGB channels as the target?

2)  Why does introducing composite color spaces improve object-centric representation? Could you provide a more detailed explanation from a feature perspective?

3) How does the model behave for different values of the number of slots? Is there an ablation showing the effect of that parameter?

---

> ### Author Response · Authors · 2024-11-27
>
> We thank the reviewer for the insightful feedback and hope our answer addresses the concerns and questions raised.
>
> "The authors only discuss grayscale for the single-channel approach. Considering the high correlation between RGB channels, what would the impact for OCRL be if using just one of the RGB channels as the target?"
>
> We implemented experiments to investigate this effect and added training on the "R" channel of RGB to our analysis in Section 3. We observe moderately substantial differences between training on the single R channel and RGB. The single R channel suffices for the Clevr dataset but lacks performance on the more challenging Clevrtex dataset.
>
> "Why does introducing composite color spaces improve object-centric representation? Could you provide a more detailed explanation from a feature perspective?"
>
> We argued in Section 3 about the drawbacks of the RGB space. For example, there is a high correlation between lightness and only subtle differences between the color channels representing hue and saturation.
> Regarding the effect of composite color spaces on the observed object-centric representation improvements, we could not identify a single detailed reason - We investigated the effect and potential reasons in more detail in Appendix D4, where we explore the impact of composite color spaces for scenes under different lighting conditions. We observed that slots obtained from RGB-S targets tend to be more robust against lighting changes, which can be one effect. We furthermore highlight the attention maps of slots in Appendix D5. Slots trained on the RGB space tend to occupy fixed spatial areas instead of objects. We suppose this happens because spatial areas often share similar lighting conditions, which are overrepresented in the RGB space - the hue and saturation channels of the HSV space are robust to lighting, helping slots in discriminating objects from the background.
>
> "How does the model behave for different values of the number of slots? Is there an ablation showing the effect of that parameter?"
>
> We added an ablation for the Clevrtex dataset in the Appendix, Section D.7.  Although models trained on RGB do not degenerate as often with fewer slots, the RGB-S models still outperform them consistently - the only exception is for a very low number of slots (5).
>
> "Although the authors state that their work focuses on color spaces, the essence of their approach lies in enhancing the model's perceptual abilities by adjusting target objectives. Therefore, using only an RGB-RGB baseline may not be reliable. It would be more robust to include comparisons with methods that use different representations as targets, such as (Seitzer et al., 2023)."
>
> We agree with the Reviewer that a comparison to other target signals, e.g., self-supervised objectives, is interesting. We thus added a comparison to the Dinosaur [1] method in the Appendix, Section D.6. Our preliminary results show that Dinosaur shows superior performance on Movi-C but lacks performance on Clevrtex. Furthermore, we argue in the appendix that self-supervised signals show a massive improvement for real-world datasets, but the corresponding self-supervised signal also limits their performance. Contrarily, the composite RGB-S spaces contain all information and might thus be combined with self-supervised signals. For example, one could predict the composite color spaces as additional targets to self-supervised signals.
>
> "While the authors' idea is interesting, they do not provide a reliable analysis from a representation perspective on the inclusion of different color space targets. For instance, in Figure 2 of the main paper, there is a noticeable similarity between saturation and the mask in highlighting foreground objects; however, the mask is discrete and contains categorical information, while saturation focuses solely on object shape. There is insufficient technical analysis explaining the impact of using saturation."
>
> We ask the Reviewer to clarify this point. We do not claim that saturation acts similarly to the object masks. However, we claim that the saturation contains complementary features that might be used to discriminate between foreground and background objects, which are not well represented in the RGB space.
>
> [1] M. Seitzer, M. Horn, A. Zadaianchuk, D. Zietlow, T. Xiao, C.-J. Simon-Gabriel, T. He, Z. Zhang, B. Schölkopf, T. Brox, and F. Locatello. Bridging the Gap to Real-World Object-Centric Learning, March 2023. URL http://arxiv.org/abs/2209.14860. arXiv:2209.14860 [cs].

---

> > ### Comment · Reviewer_puBu · 2024-11-28
> >
> > Thanks for the authors' efforts and feedback. However, there are still some questions.
> >
> > 1) As shown in Fig.6, if high variance means better "attending to objects close to that area", the results of HSV seem better than RGB-S (consistent with the results shown in Tab.6). But, it is confused about the results of Tab.2 and it is still unclear about the property of composite color space
> >
> > 2) Could you please provide intermediate feature maps rather than attention maps to show the influence of different color spaces, which can better describe the difference among different color spaces?
> >
> > 3)For Fig.2, it only shows the results of RGB-HSV reconstruction, could you provide the comparisons between RGB-RGB and RGB-HSV to better explain the possible "binding" issues?
> >
> > I hope the authors can resolve my concerns.

---

> > > ### Author Response · Authors · 2024-11-28
> > >
> > > We thank you for your very fast reply and the factual follow-up questions.
> > > We have incorporated your concerns in a new version of the PDF uploaded just now and hope to address your concerns there and in the following:
> > >
> > > 1. We agree with the first statement. The confusion results from a misunderstanding with a caveat (that was stated in the Figure 6 caption) but could be made more explicit. We, therefore, clarified this in the Figure caption. Figure 6 only shows the attention maps for ClevrTex, where HSV performs indeed better than RGB-S. This is also consistent with Table 6 (with FG-ARI  and mIoU being higher for HSV), but it is also consistent with the specific part about ClevrTex in Tab. 2, where HSV is indeed performing better than RGB-S. Overall, however, this is not true for the other datasets, and Figure 6 would look different for other evaluation datasets, where RGB-S would have a higher variance than HSV. This hopefully solves the confusion: The results in Figure 6 apply only to ClevrTex, the only dataset where HSV is better than RGB-S, which is why we investigated it in more detail. The results overall are consistent with this for the color spaces.
> > >
> > > 2. We can. We are currently working on generating those. We hope we can upload these here, if allowed and technically possible, asap.
> > >
> > > 3. There is some additional information on this in the Appendix. We clarified this in the caption for Figure 2 to point to the Appendix (in particular Figure 9 and others), where the described binding effect can be observed in detail by comparing all target color spaces and datasets as requested. We see in Figure 9 that color spaces, including the hue channel, degenerate for the Clevr test set to bind areas instead of objects, while all other color spaces bind objects. For other test sets in Figure 10+, we observe varying results, which are discussed in the main text in the quantitative evaluation parts, see Section 5.1).

---

> > > > ### Author Response · Authors · 2024-12-02
> > > >
> > > > Once again, we thank the reviewer for his constructive feedback and questions. We have now uploaded the feature maps at the anonym link:
> > > >
> > > > https://github.com/ocrlAnonym/iclr_2025_slot
> > > >
> > > > At this location, there are folders for two datasets:
> > > >
> > > > - Clevr (folder Clevr)
> > > > - Clevrtex (folder Clevrtex)
> > > >
> > > > For each dataset, we present five samples (5 subfolders), where we compare the RGB, RGB-S, and HSV feature maps. For quick reference, the root directory contains an overview figure, with the higher resolution raw figures in each subdirectory, e.g., "clevr_0" for the first sample of the clevr dataset. Note that those are only the feature maps of the encoder, as Slot Attention utilizes a spatial broadcast decoder for each slot. In each subfolder, the following files can be found:
> > > > - "clevr_0_image.png" for the original image
> > > > - "clevr_0_RGB_mask.png" for the masks of the color spaces. The HSV and RGB-S masks are also provided
> > > > - "clevr_0_gt_mask.png" for the ground truth mask of the scenes
> > > > - "clevr_0_HSV - Layer_1.png" for the feature map of layer one on the HSV space.
> > > >
> > > > For most images, we can notice slightly improved spatial separation for the detected objects (e.g., compare layer 4 for RGB and RGB-S for "clevrtex_3"). The color spaces directly influence the learned binding mechanism of the Slot Attention module.

---

> > > > > ### Comment · Reviewer_puBu · 2024-12-02
> > > > >
> > > > > Thanks for your feature maps. Totally, I think this paper well show and analyze the effects of composite color space for object-centric representation learning. Combining other reviewers' feedbacks, I would like to increase the score to 6. However, in the future, I hope authors can provide a deeper insight for this phenomenon, not limited to specific models or colors.

---

### Official Review · Reviewer_hEfG · 2024-11-05

**Soundness:** 2
**Presentation:** 2
**Contribution:** 2
**Rating:** 3
**Confidence:** 3

**Summary:**

This paper explores the influence of color channels on object-centric representation learning (OCRL). It first shows the strong correlation, sensitivity to lighting, and non-uniformity of the RGB space. Then a combination of RGB with other color channels is studied for OCRL. Experimental results show that RGB with the saturation channel in HSV achieves consistent improvement on all datasets used in this paper. The paper is too technical and lacks novelty as the combination of color space is often used as a trick in computer vision and image processing tasks.

**Strengths:**

1. Give a detailed analysis of RGB space.
2. Verify that RGB with S channel achieves consistent performance gain on different datasets.

**Weaknesses:**

1. The paper is not easy to read as the cross-referencing of figures and tables is disordered.
2. It claims that "We challenge the common assumption that RGB images are the optimal target for unsupervised learning in computer vision and demonstrate this for OCRL". It's better to give a reference which shows the RGB image is optimal.
3. Table 2 is not too strong to support that RGB-S is good enough. Mostly, it's not the best on different datasets, for example, RGB-SV is better than RGB-S on Clevr, MultishapesNet-4, and MultishapesNet-24; HSV is the best for Clevrtex.
4. The experimental analysis is not convincing, for example in Line 457, which one is most important for representation, the color space or a powerful decoder?
5. How does the model guarantee universality when it is trained on the combined color space?  I still think RGB is the most universal space even though the performance is slightly worse than others in Table 2.

**Questions:**

See weakness above.
The biggest concern for me is the novelty of the paper and the universality of the model trained on various numbers of color channels.

---

> ### Author Response · Authors · 2024-11-27
>
> We thank the reviewer for their comments and helpful suggestions. In the following section, we will address the concerns and questions raised.
>
> - "The paper is not easy to read as the cross-referencing of figures and tables is disordered."
>
>     We changed the placement of figures in the main text to appear next to their reference in the text. Specifically, we adjusted the positions of Figure 1, Figure 2, and Figure 4. The tables are positioned next to their reference in the main text.
>
> - "It claims that "We challenge the common assumption that RGB images are the optimal target for unsupervised learning in computer vision and demonstrate this for OCRL". It's better to give a reference which shows the RGB image is optimal."
>
>     We agree with the reviewer and changed the sentence to "We challenge the common assumption that RGB images are the optimal color space for unsupervised learning in computer vision and demonstrate this for OCRL." Furthermore, we added references to SOTA unsupervised computer vision networks, such as CLIP, DINO, and Slot Attention.
>
> - "Table 2 is not too strong to support that RGB-S is good enough. Mostly, it's not the best on different datasets, for example, RGB-SV is better than RGB-S on Clevr, MultishapesNet-4, and MultishapesNet-24; HSV is the best for Clevrtex."
>
>     We agree with the reviewer that the RGB-S space is not the best-performing color space for all considered datasets, as stated. We toned down potentially ambiguous claims in the main text in lines 334 and 538 to clarify that other composite color spaces next to RGB-S can be similarly helpful and that the RGB-S and RGB-SV space consistently improve the performance over the current color space used for OCLR (RGB).
>
> - "The experimental analysis is not convincing, for example in Line 457, which one is most important for representation, the color space or a powerful decoder?"
>
>     We thank the reviewer for pointing out this unclarity in our writing. Table 2 shows that the composite color spaces improve the performance of OCLR networks independent of the model's complexity. We changed our main text (line 415) for clarity.
>
> - "How does the model guarantee universality when it is trained on the combined color space? I still think RGB is the most universal space even though the performance is slightly worse than others in Table 2."
>
>     There might be a misunderstanding. We do not aim to claim that the model guarantees universality. We argue that models trained on combined color spaces are less dependent on inherent biases of the color spaces and evaluate this effect statistically. We added this to our main text in Section 3.3 to avoid confusion. For example, differences in hue and saturation are subtle in the RGB space (due to the high correlation with the lightness). We countersteer this effect by combining multiple color spaces - for example, adding the hue and saturation to the RGB color space pronounces those attributes more. Regarding the claim that RGB is still the most universal color space, we find such a claim difficult, also based on the physics behind color itself (light spectrum) and the somewhat arbitrary, biologically-induced choice for three channels. Also, such a claim would then transfer to some other color spaces as well, such as HSV and LAB, as they can be losslessly induced from the RGB space, then claimed to be  as universal - but we still see different statically evaluation results for them. Nevertheless, we agree with the Reviewer that RGB is the most popular color space in computer vision and widely used.
>
> - "The paper is too technical and lacks novelty as the combination of color space is often used as a trick in computer vision and image processing tasks."
>
>    We agree with the Reviewer that changing the color space is a common trick in computer vision and is already well explored and tested. Nevertheless, we contribute two novelties, going beyond the tricks and computer vision contributions referenced in the related work section: Firstly, we combine color spaces and theoretically and empirically reason about its advantages. Secondly, unlike related work, we utilize these color spaces in unsupervised OCLR as a target instead of the input, showing improvements above the SOTA. We state this more explicitly in our contribution section. Those two contributions set apart our work from previous research.

---

> > ### Comment · Reviewer_hEfG · 2024-11-28
> >
> > Thanks for the authors' feedback. Based on the rebuttal, I still think the proposed method is too tricky and lacks novelty. I keep to the score of reject.

---

> > > ### Author Response · Authors · 2024-11-28
> > >
> > > We appreciate your follow-up comment and thank you for further clarifying your perspective. While we understand and respect your viewpoint, we would like to offer a final clarification on the contributions and positioning of our work.
> > >
> > > 1. Novelty: We acknowledge your concern regarding novelty and reiterate that our contributions go beyond merely combining color spaces, as stated in our rebuttal. Specifically, our work:
> > >     - Provides a novel theoretical and empirical examination of how combined color spaces can mitigate inherent biases in commonly used color representations (e.g., RGB).
> > >     - Explores the application of these combined spaces in unsupervised object-centric representation learning (OCRL), where color space manipulation is typically not explored in the context of unsupervised learning targets. These results highlight a pathway for improving state-of-the-art OCRL models, which we believe offers substantial value to the field.
> > > 2. Practical Impact: While the combination of color spaces may appear as a "trick," we emphasize that our analysis rigorously demonstrates how these combinations lead to consistent performance improvements across diverse datasets. This consistency underscores the practical utility of our method in real-world OCRL applications beyond its theoretical implications.
> > >
> > > 3. Universality vs. Practicality: We understand that RGB is a widely accepted standard in computer vision, and our work does not aim to displace it. Instead, we propose that incorporating additional color channels can reduce bias and improve task-specific performance. This perspective complements, rather than contradicts, the universality of RGB. We respect your assessment and thank you for the constructive feedback that has helped us to refine our manuscript. While we regret that our work did not fully align with your expectations of novelty, we hope the clarified contributions are valuable to the broader community.
> > >
> > >
> > > May we finally ask you to provide us with some references that underscore your assessment that our approach lacks novelty to better understand your concerns, so we can fully understand what we can do to make such a contribution novel and less tricky in your assessment?

---

### Author Response · Authors · 2024-11-27
**Adressed reviewer comments for 11349:  Leveraging Color Channel Independence for Improved Unsupervised Object Detection**

We thank the reviewers for their detailed, constructive, and helpful feedback and their valuable insights. In the following, we describe the changes we made to the paper. We have aimed to address all concerns raised and hope to clarify some ambiguities. For the revised paper, please note all changes are marked in red in both (1) the main text as well as (2) the appendix, which includes a new and extended set of experiments. Specifically, we added experiments extending the analysis of the influence of composite color spaces on slot representations (Appendix D.4 and D.5), a comparison to self-supervised methods (Appendix D.6), and an ablation study of the number of slots (Appendix D.7).

Primarily, we focused on (1) adding additional experiments - as requested - to back up all claims made (main text: key facts, details in appendix), (2) sharing additional insights, intuition, and further reasoning, based on existing as well as additional experiments and foundations in related work (both in main text and appendix) to address issues raised by the reviewers, and (3) improve reasoning and clarify concerns raised, further enhancing the specific impact, scope, and contribution descriptions (main text).


We are confident that these changes improve the quality, reproducibility, and understandability of the paper significantly thanks to the reviewers' feedback, helping to clarify our findings and address their raised concerns fully - in particular, as there was some uncertainty about the correct understanding of some parts.
Again, we thank the reviewers for their hard work and are open to further feedback and questions.

---

### Note · Authors · 2024-12-13

I have read and agree with the venue's withdrawal policy on behalf of myself and my co-authors.